# CLoSeR: Continual Learning in VQ-GAN for Test-time Style Refinement

## Abstract

While existing artistic style transfer methods enable cross-domain image synthesis, they often struggle to strike a balance among stylistic realism, inference efficiency, and geometric consistency. *To address this limitation, we propose a test-time refinement (TTR) framework that universally enhances stylistic fidelity through a scalable and self-supervised VQ-GAN refiner, while keeping the parameters of the underlying style-transfer generator frozen.* Our primary contribution is a continual learning framework for VQ-GAN, which combines *Low-Rank Adaptation* (LoRA) with incremental codebook expansion. This design enables efficient adaptation to diverse artistic styles while preserving previously learned knowledge, significantly reducing the computational and memory overhead of deploying models across multiple domains. Notably, our approach reduces the number of trainable parameters by up to 94% compared to full-model fine-tuning, offering a highly parameter-efficient solution for test-time refinement. Furthermore, we introduce positional embeddings into the latent embedding space, which strengthens the model's geometry awareness and improves structural coherence in the generated results. We name our approach CLoSeR (*Continual Learning in VQ-GAN for Style Refinement*), and evaluate it across multiple style transfer benchmarks under a test-time adaptation setting. Experimental results show that CLoSeR improves style fidelity and structural consistency, achieving a maximum relative reduction of 44% in *Fréchet Inception Distance* (FID), demonstrating significant gains in generation quality. The code will be released.

## 1 Introduction

*Artistic style transfer* (AST) has witnessed rapid progress through a variety of approaches, most notably neural style transfer (NST) (Gatys et al., 2016; Huang & Belongie, 2017; Liu et al., 2021; Hong et al., 2023) and generative adversarial networks (GANs) (He et al., 2018; Lee et al., 2020; Huang et al., 2024). These methods typically rely on one or a few reference style images to guide the stylization process. More recently, diffusion models (Zhang et al., 2023; Chung et al., 2024; Wang et al., 2024; Zhou et al., 2025), autoregressive (AR) approaches (Li et al., 2024), and flow-based generative methods (Lipman et al., 2022; Geng et al., 2025) have demonstrated impressive capabilities in producing high-quality and diverse stylizations, often supporting multimodal inputs. These advances highlight the growing importance of transferable representations that capture both content and stylistic priors, enabling more flexible and controllable AST.

However, existing methods struggle to achieve an optimal balance between content consistency, stylistic realism, and inference efficiency. NST and GAN-based methods (Gatys et al., 2017; Selim et al., 2016; Zhu et al., 2017) enable fast inference and preserve geometric structure well, but often fail to learn sufficiently rich representations of artistic textures. Diffusion models (Zhang et al., 2023; Wang et al., 2024; Ye et al., 2025) generate high-quality results with nuanced style patterns, yet suffer from hallucinated content, weak content–style correspondence, and the high computational cost due to iterative sampling. Reducing inference steps typically degrades image quality significantly. Moreover, both diffusion and AR models often yield over-smoothed textures, suggesting that their learned representations do not fully align with the expressive nature of real-world artistic styles. Few-shot or training-free adaptation methods (Chung et al., 2024; Farhadzadeh et al., 2025) further face challenges in building robust representations for unseen domains. Thus, learning domain-aligned and structurally consistent representations remains an open challenge for AST.

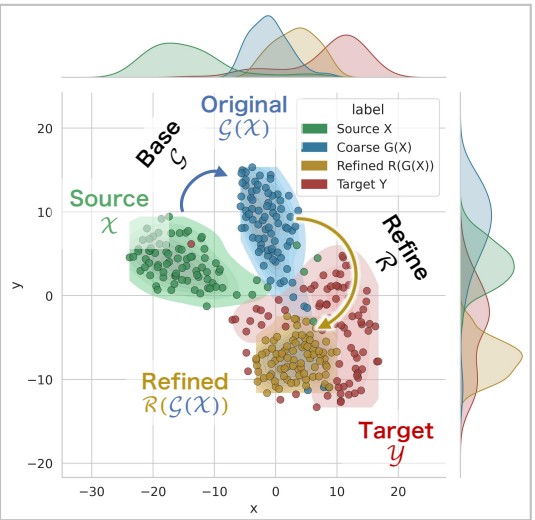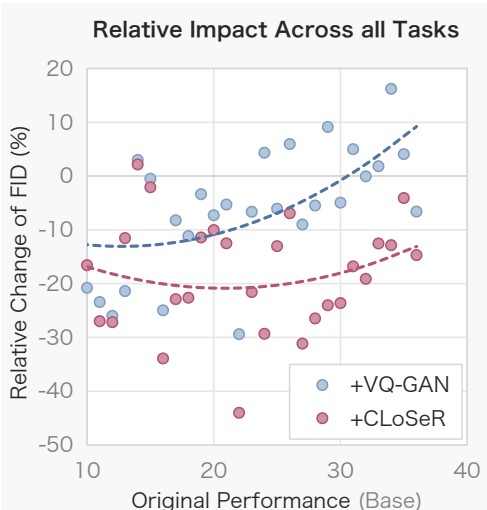

Figure 1: Motivation of CLoSeR. **Left:** illustration of the distribution shift from the source domain (CelebAMask-HQ (Lee et al., 2020)) to the target domain (MetFace (Karras et al., 2020)). StyleID (Chung et al., 2024) serves as the base model to generate coarse outputs, while our CLoSeR produces refined results that align more closely with the target domain. Features are extracted with VGG-19 (Simonyan & Zisserman, 2014) and visualized via t-SNE.) **Right:** scatter plot of refined performance versus original performance across diverse base models and artistic styles. Lower FID values indicate better style fidelity.

As illustrated in Fig. 1, the motivation for our approach stems from the persistent distributional gap between stylized outputs and the target domain. While existing image translation models—such as GAN-, attention-, and diffusion-based methods (Huang & Belongie, 2017; Liu et al., 2021; Yi et al., 2019; Chung et al., 2024; Zhou et al., 2025) —can roughly map source content into the target style space, their outputs often exhibit significant deviations from the authentic target distribution, particularly in terms of stylistic fidelity and geometric consistency (left panel). These gaps indicate a representation mismatch between the generated outputs and the target artistic domain.

To address this, we explore an alternative perspective: rather than retraining or modifying the generator, we refine its outputs at test time through reconstruction in the embedding space. Inspired by the ability of VQ-GAN (Esser et al., 2021) to learn a compact, self-supervised representation of the target domain, we propose a *test-time refinement* (TTR) framework that leverages VQ-GAN as a domain anchor. In other words, VQ-GAN refines coarse stylized images by aligning their features with a pre-learned target domain representation in its latent codebook, eliminating the need for generator updates.

However, directly fine-tuning VQ-GAN for each new style remains computationally expensive and lacks scalability. To overcome these limitations, we propose a TTR framework dubbed **CLoSeR**, *i.e. Continual Learning in VQ-GAN for Style Refinement*. CLoSeR enables efficient continual adaptation by incrementally enriching the learned representation space through *Low-Rank Adaptation* (LoRA) (Hu et al., 2022) and codebook expansion. This design drastically reduces the number of trainable parameters—by over 94% compared to full fine-tuning—while preserving previously acquired representations of earlier styles. Furthermore, to mitigate structural distortions caused by the lack of spatial awareness in vanilla VQ-GAN (Esser et al., 2021), we incorporate 2D sine-cosine positional embeddings (Vaswani et al., 2017; Carion et al., 2020) into the latent representation space, endowing the codebook and decoder with explicit spatial priors. Together, these components enable CLoSeR to refine generation quality through representation learning, achieving both high-fidelity stylization and geometric consistency across diverse artistic domains.

We conduct extensive experiments to evaluate the effectiveness and generality of our approach. The results demonstrate that CLoSeR consistently improves generation quality across diverse style transfer pipelines—including GAN- (Yi et al., 2019; Zhang et al., 2022), attention- (Liu et al., 2021;

Hong et al., 2023), and diffusion-based (Kwon & Ye, 2022; Chung et al., 2024; Zhou et al., 2025) models—under both single-style and continual learning settings. The framework enhances stylistic realism and structural consistency, while also learning transferable representations. As shown in the right panel of Fig. 1, a scatter plot of FID improvement reveals that both the baseline VQ-GAN and CLoSeR reduce stylization errors compared to the original outputs, but CLoSeR achieves significantly greater FID reduction, particularly in challenging cases with higher baseline errors. This confirms its superior refinement capability and scalability in real-world deployment scenarios.

## 2   RELATED WORKS

**Artistic Style Transfer.**   Early approaches leveraged CNNs to decouple style and content representations, enabling stylized image synthesis (Gatys et al., 2016; Johnson et al., 2016; Jing et al., 2019). Subsequent methods aimed to enhance style diversity and generalization by introducing adaptive normalization and attention-based mechanisms (Huang & Belongie, 2017; Park & Lee, 2019; Hong et al., 2023). More recently, diffusion-based approaches have emerged as powerful alternatives for style and domain transfer (Ho et al., 2020; Kwon & Ye, 2022; Gu et al., 2022). These methods have been applied to stylization, latent space disentanglement, and domain adaptation by exploiting denoising priors and structured noise injection (Kwon & Ye, 2022; Su et al., 2022; Parmar et al., 2024; Zhou et al., 2025). In parallel, large pretrained text-to-image (T2I) diffusion models have been adapted to AST, enabling prompt-driven any-to-any stylization without case-by-case retraining (Rombach et al., 2022; Brooks et al., 2023; Chen et al., 2023). In addition, training-free paradigms have been explored to achieve lightweight and interpretable transfer (Chung et al., 2024). Despite these advances, both CNN-based and diffusion-based pipelines often struggle with preserving structure and maintaining style fidelity in complex artistic domains.

**Vector Quantization.**   Vector Quantization (VQ) has emerged as a powerful technique for learning discrete representations. VQ-VAE (Van Den Oord et al., 2017) pioneered vector quantization in generative modeling, and VQ-GAN (Esser et al., 2021) further advanced this direction. Building on the success of VQ-GAN, a variety of works have emerged, such as VQ-Diffusion (Gu et al., 2022) for text-to-image generation and QuantArt (Huang et al., 2023) for artistic style transfer. Reconstruction and generation using VQ have also been widely studied (Zhu et al., 2024; Yu et al., 2024; Yao et al., 2025). In the autoregressive paradigm, Li et al. (2024) propose eliminating discrete quantization entirely by modeling per-token distributions, while MergeVQ (Li et al., 2025) unifies representation learning and generation through token merging and a lookup-free quantization strategy.

**Continual Learning.**   Continual learning has been extensively studied, but its application to artistic domains remains relatively underexplored. Traditional style transfer methods often require retraining for each new style (Gatys et al., 2016; Johnson et al., 2016), making them inefficient and vulnerable to catastrophic forgetting. To address these limitations, modular and parameter-efficient approaches have been proposed (Liang & Li, 2024; Zhu et al., 2025; He et al., 2025; Roy et al., 2023). More recently, continual generative learning has incorporated strategies such as replay (Caccia et al., 2020; Jeon et al., 2023), distillation (Lesort et al., 2019; Zhao et al., 2020), and modularization (Yoon et al., 2018). LoRA-based adapters (Hu et al., 2022; Farhadzadeh et al., 2025) have proven particularly effective, enabling lightweight, style-specific modules to be integrated into frozen backbones for scalable, efficient, and largely forget-free adaptation. However, they still suffer from increasing knowledge degradation as the number of tasks grows (Liang & Li, 2024).

## 3   METHOD

### 3.1   OVERVIEW

We propose CLoSeR (*Continual **L**earning in VQ-GAN for **S**tyl**e** **R**efinement*), a *test-time refinement* (TTR) framework that enhances both stylistic realism and geometric consistency in artistic style transfer. The pipeline of CLoSeR is shown in Fig. 2. Building upon VQ-GAN (Esser et al., 2021), we integrate parameter-efficient adaptation through *Low-Rank Adaptation* (LoRA) and incremental codebook expansion, supporting continual adaptation to new styles with minimal overhead. For each new style, only a lightweight LoRA module and a style-specific discriminator are trained,

Figure 2: Overview of **CLoSeR**, *i.e.*, *Continual Learning in VQ-GAN for test-time Style Refinement*. (a) New styles are integrated by expanding the codebook ($\mathcal{C}^{(i)}$) while retaining the base style representation $\mathcal{C}_0$ (style 0). The encoder features are enriched with cosine-sine positional embeddings and reconstructed by the decoder with LoRA-based adaptation. (b) Given an initial coarse stylized output $\mathcal{G}(x)$ from any generator, CLoSeR reconstructs it through the learned codebook, aligning the result with the target style domain.

while the shared VQ-GAN backbone remains frozen. This strategy enables scalable deployment in dynamic and long-tail style scenarios. In addition, our approach introduces *geometry-aware vector quantization* by embedding positional encodings into the latent space, allowing the model to incorporate explicit spatial priors during reconstruction and thereby correcting geometric distortions and local artifacts commonly present in coarse stylized outputs. Finally, CLoSeR operates in a *plug-and-play* manner and can be applied to enhance outputs from arbitrary generative models.

## 3.2 CONTINUAL LEARNING IN VQ-GAN VIA LoRA

Adapting to new artistic styles while preserving previously learned knowledge remains challenging due to catastrophic forgetting and the large parameter overhead of full fine-tuning. To enable efficient and scalable continual learning, we integrate LoRA (Hu et al., 2022) and incremental codebook expansion into the VQ-GAN framework, allowing CLoSeR to adapt to new styles with minimal trainable parameters while keeping the shared backbone frozen.

**LoRA-based Encoder–Decoder Adaptation.** We apply LoRA to all convolutional layers of the encoder and decoder, injecting trainable low-rank matrices to modulate features in a style-specific manner. Specifically, each pre-trained weight $W_0 \in \mathbb{R}^{d \times k}$ is updated as:

$$W = W_0 + \frac{\alpha}{r} AB, \quad A \in \mathbb{R}^{d \times r}, \ B \in \mathbb{R}^{r \times k}, \tag{1}$$

where $A$ and $B$ are the low-rank adaptation matrices. $A$ is initialized with zeros, $B$ with a standard normal distribution, $\alpha$ is a scaling factor, and $r$ is the rank (set to 8 in our experiments). The original weights $W_0$ remain frozen and are shared across all styles.

**Incremental Codebook Expansion.** For each new style $s_i$, we expand the codebook with $\Delta K = 1024$ additional entries:

$$\mathcal{C}^{(i)} = \mathcal{C}_0 \cup e_{K_0+1}, \ldots, e_{K_0+\Delta K}, \tag{2}$$

where $\mathcal{C}_0$ denotes the initial codebook. This strategy enables the model to encode style-specific visual primitives while preserving previously learned representations.

**Training.** During training on style $s_i$, only three components are updated: the LoRA parameters $\Theta_{\text{LoRA}}^{(i)}$, the newly added codebook entries $\mathcal{C}^{(i)} \setminus \mathcal{C}_0$, and a lightweight style-specific discriminator $\mathcal{D}^{(i)}$. All other parameters—including the encoder, decoder, and the base codebook—remain frozen.

**Inference.** At inference, given an initial stylized result $\mathcal{G}(x)$ from any pre-trained generator, the refined output for style $s_i$ is computed as:

$$\hat{y} = \mathcal{R}\big(\mathcal{G}(x); \Theta^{(i)}\big), \quad \text{with} \quad \Theta^{(i)} = \Theta_{\text{LoRA}}^{(i)}, \mathcal{D}^{(i)}. \tag{3}$$

This modular design enables plug-and-play refinement: users select the target style, and the system loads the corresponding lightweight parameters, thereby avoiding redundant computation and supporting efficient deployment in dynamic or long-tail scenarios.

### 3.3 GEOMETRY-AWARE VQ-GAN

To improve spatial structure preservation in artistic style reconstruction, we enhance the VQ-GAN framework (Esser et al., 2021) with 2D sine–cosine positional embeddings injected into the latent representation space. Unlike standard VQ-GAN, which processes latent features without explicit spatial inductive bias, our method embeds positional priors prior to quantization—thereby enabling geometry-aware refinement without introducing any additional learnable parameters.

Similar to Transformer (Vaswani et al., 2017), for each spatial position $(m, n) \in \{1, \ldots, h\} \times \{1, \ldots, w\}$ of the continuous latent feature map $f_s \in \mathbb{R}^{h \times w \times d}$, we generate a corresponding 2D positional embedding $P_{m,n} \in \mathbb{R}^d$ using an extended sine-cosine scheme:

$$P_{m,2i} = \sin\left(\frac{m}{10000^{\frac{2i}{d}}}\right), \quad P_{m,2i+1} = \cos\left(\frac{m}{10000^{\frac{2i}{d}}}\right),$$
$$P_{n,2i} = \sin\left(\frac{n}{10000^{\frac{2i}{d}}}\right), \quad P_{n,2i+1} = \cos\left(\frac{n}{10000^{\frac{2i}{d}}}\right), \tag{4}$$

where $m$ and $n$ denote the row and column indices, $i$ is the dimension index, and $d$ is the embedding dimension. The positional embedding $P_{m,n}$ is then added element-wise to the latent feature $f_s$:

$$f_{pe} = f_s + P_{m,n}, \tag{5}$$

forming spatially enriched features that retain semantics and explicit structure.

The enhanced features $f_{pe}$ are then passed to the codebook for quantization:

$$Q_{\mathcal{C}}(f_{pe}) := \arg\min_{\mathbf{c}_i \in \mathcal{C}} \|f_{pe} - \mathbf{c}_i\|, \tag{6}$$

where $\mathbf{c}_i$ denotes the $i$-th code vector in the codebook $\mathcal{C}$. By integrating explicit spatial priors into the vector quantization pipeline, our approach effectively improves geometric consistency in the reconstructed outputs, particularly in structure-sensitive artistic domains.

### 3.4 LOSS FUNCTIONS

To balance pixel-level fidelity, perceptual quality, quantization alignment, and adversarial realism, we adopt a composite loss composed of multiple complementary objectives.

**Reconstruction Objective.** The reconstruction objective combines an L1 pixel-wise loss and a perceptual loss in deep feature space. Given the input image $x_s$ and its reconstruction $y_s$, the pixel-level reconstruction loss is defined as $\mathcal{L}_{\text{rec}} = \|y_s - x_s\|_1$. To capture higher-level semantic consistency, we further employ the LPIPS metric (Zhang et al., 2018) as a perceptual loss:

$$\mathcal{L}_{\text{perc}} = \text{LPIPS}(x_s, y_s). \tag{7}$$

The total reconstruction loss is then given by:

$$\mathcal{L}_{\text{recon}} = \mathcal{L}_{\text{rec}} + \lambda_{\text{perc}} \cdot \mathcal{L}_{\text{perc}}, \tag{8}$$

where $\lambda_{\text{perc}}$ controls the relative weight of perceptual similarity.

**VQ Loss.** Following standard practice in vector quantized models (Esser et al., 2021), we incorporate a vector quantization (VQ) loss to align the latent space with the codebook. Let $f_s \in \mathbb{R}^{B \times C \times H \times W}$ denote the continuous latent features from the encoder. We enrich these features with 2D sine-cosine positional encoding (see Section 3.3) to obtain $f_{\text{pe}}$, which is then flattened and mapped to the nearest entries in a learnable codebook $\mathcal{C} \in \mathbb{R}^{K \times D}$, where $K$ is the number of codebook vectors and $D$ is the embedding dimension. The quantized output $z_{\text{q}}$ replaces each feature in $f_{\text{pe}}$ with its closest codebook entry under the Euclidean distance. To jointly optimize the codebook and encoder, we use the following VQ loss:

$$\mathcal{L}_{\text{VQ}} = \|\mathbf{sg}[z_{\text{q}}] - f_{pe}\|_2^2 + \beta\|\mathbf{sg}[f_{pe}] - z_{\text{q}}\|_2^2, \tag{9}$$

where $\mathbf{sg}[\cdot]$ denotes the stop-gradient operator and $\beta$ is a hyperparameter controlling the codebook update strength.

**Adversarial Loss.** For adversarial training, we adopt the standard cross-entropy objective as in VQ-GAN (Esser et al., 2021). The discriminator $\mathcal{D}^{(i)}$ for style $s_i$ is optimized as:

$$\mathcal{L}_{\text{adv}} = -\mathbb{E}[\log \mathcal{D}^{(i)}(y)] - \mathbb{E}[\log(1 - \mathcal{D}^{(i)}(y_s))], \tag{10}$$

where $y$ and $y_s$ denote real and reconstructed images.

**Total Loss.** The overall training objective is a weighted combination of all components:

$$\mathcal{L}_{\text{total}} = \mathcal{L}_{\text{recon}} + \lambda_{\text{VQ}}\mathcal{L}_{\text{VQ}} + \mathcal{L}_{\text{adv}}, \tag{11}$$

where $\lambda_{\text{VQ}}$ is set to 0.1 by default. This multi-objective formulation ensures high-fidelity, geometrically coherent, and stylistically realistic reconstructions.

# 4 EXPERIMENTS

## 4.1 SETTINGS

**Datasets & Metrics.** For the **Artistic Portrait** domain, we use MetFace (Karras et al., 2020), APDrawing (Yi et al., 2019), and FS2K (Fan et al., 2022) as style datasets, with facial photos from CelebAMask-HQ (Lee et al., 2020) and FS2K serving as content images. For the **Natural Scene** domain, we collect data from Flickr and WikiArt. We adopt standard metrics—ArtFID (Wright & Ommer, 2022), FID (Heusel et al., 2017), and KID (Bińkowski et al., 2018)—to quantitatively evaluate our results. All images are resized to $256 \times 256$ before training and evaluation.

**Implementation Details.** Following the architecture design of QuantArt (Huang et al., 2023), the encoder and decoder each consist of four blocks, with two ResBlocks (He et al., 2016) and a down-sampling/upsampling layer. The quantized feature map has a spatial resolution of $16 \times 16$ and an embedding dimension of 256. The codebook contains $N = 1024$ entries, each of dimension $d = 256$. For training, we set the batch size to 8 and the momentum queue length to 1024. For each newly added style, the codebook is expanded by 1024 tokens. We use the Adam optimizer (Adam et al., 2014) with a learning rate of $4.5 \times 10^{-6}$. Our CLoSeR framework is implemented in PyTorch (Paszke et al., 2019), and all experiments are conducted on a single NVIDIA RTX 4090 GPU.

**Baseline Models.** We evaluate our method against a set of state-of-the-art methods, including neural style transfer (QuantArt (Huang et al., 2023), AesPA-Net (Hong et al., 2023), CAST (Zhang et al., 2022), AdaAttN (Liu et al., 2021)), and diffusion-based stylized image generation (DiffuseIT (Kwon & Ye, 2022), InST (Zhang et al., 2023), StyleID (Chung et al., 2024) and AttenDistill (Zhou et al., 2025)). For fair comparison, we use publicly available implementations with their recommended configurations. As shown in Figure 4, our method outperforms all base models in both stylization fidelity and semantic consistency. Note that APDrawingGAN (Yi et al., 2019) is specialized for pen drawings, thus we evaluate it only in its intended settings to ensure fairness.

## 4.2 PERFORMANCE EVALUATION

### 4.2.1 NATURAL SCENE STYLE TRANSFER

Unlike the standard style transfer task, we train our model to reconstruct the input and use this to refine the results of artistic style transfer results. The model is first trained on the Monet dataset and then continually extended to Van Gogh and Ukiyo-e, enabling progressive refinement across multiple styles. Experimental results demonstrate the effectiveness of our approach.

**Quantitative Analysis.** As illustrated in Fig. 3, for both Monet and Van Gogh, the average values of all three evaluation metrics consistently decrease after the initial refinement with VQ-GAN and are further reduced when applying our proposed CLoSeR. Notably, across all baselines, our method achieves substantial improvements: FID is reduced by approximately 25% on Monet and Van Gogh, KID drops by more than 30%, and ArtFID decreases by over 20%.

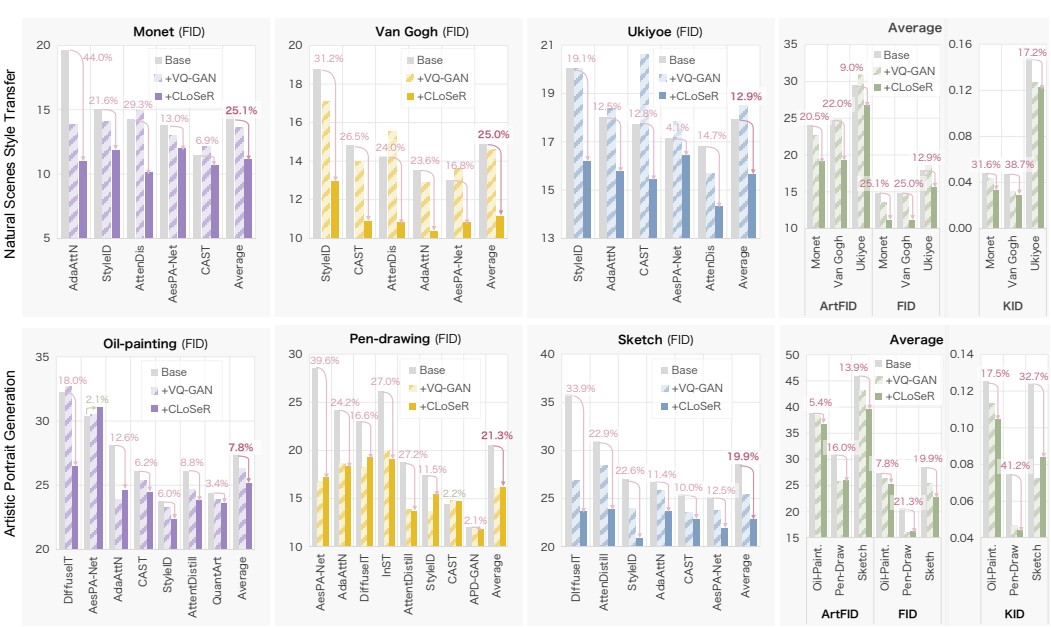

Figure 3: Quantitative performance on artistic style transfer for natural scenes and facial portraits.

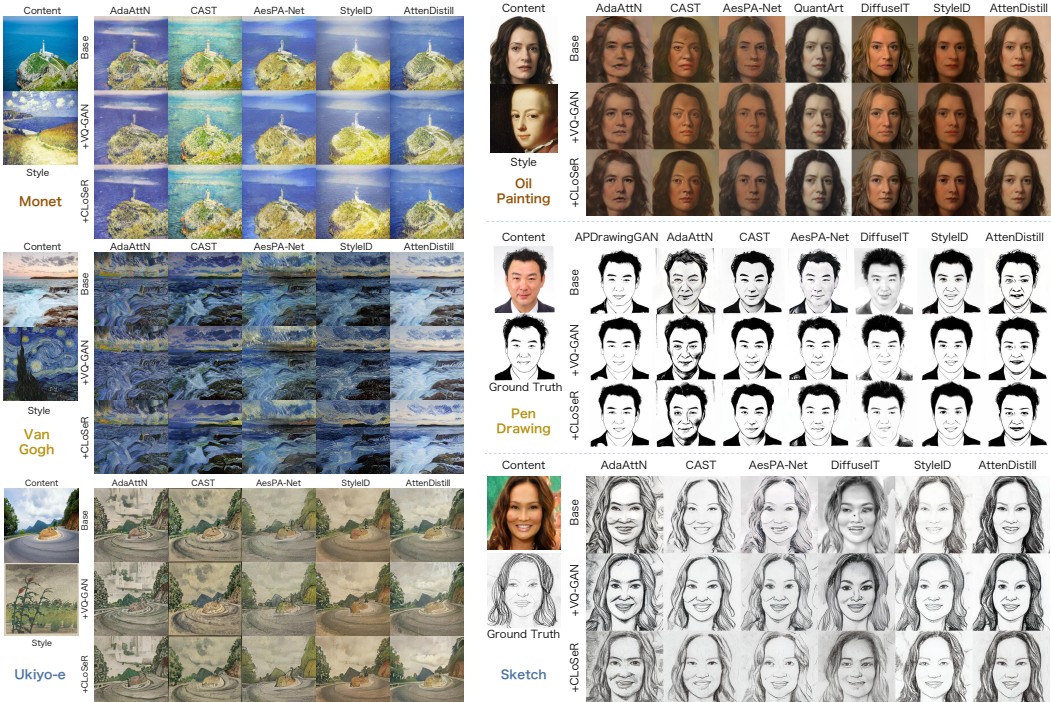

Figure 4: Generated results of different artistic styles for natural scenes and facial portraits. Please zoom in for details.

**Qualitative Analysis.** As shown in Fig. 4, CLoSeR enhances base models by recovering structural details and enriching textures. Without refinement, AdaAttN and AesPA-Net tend to produce over-smoothed outputs, while vanilla VQ-GAN introduces texture but often causes distortions. In contrast, CLoSeR yields more faithful style expression—Monet's color gradients appear smoother, Van Gogh's bold strokes are better preserved, and Ukiyo-e's flat shading and outlines remain more coherent—demonstrating improved style fidelity and content stability across diverse models.

### 4.2.2 ARTISTIC PORTRAIT GENERATION

We first pre-train CLoSeR on the MetFace (Karras et al., 2020) dataset to learn robust facial representations and extend the model to support continual refinement across two additional styles: AP-Drawing (Yi et al., 2019) and FS2K (Fan et al., 2022), resulting in a three-style refinement setup.

**Quantitative Analysis.** As shown in Fig. 3, CLoSeR consistently improves all metrics across artistic domains. Relative to the base models, it reduces FID by 7.8%, 21.3%, and 19.9% on oil painting, pen drawing, and sketch, respectively, and consistently outperforms the intermediate VQ-GAN refinement. On oil painting, CLoSeR further lowers ArtFID and KID by 5.4% and 17.5%, while ArtFID on pen drawing drops by 16.0%, indicating enhanced structural fidelity and stylistic realism across datasets and backbones.

**Qualitative Analysis.** For Oil Paintings, AdaAttN and AesPA-Net produce over-smoothed or distorted faces, while VQ-GAN reduces artifacts but suffers from leakage and color shifts. CLoSeR better preserves identity (sharper jawlines, clearer eyes) and renders textures closer to the target style. For Pen Drawings, DiffuseIT and AesPA-Net often yield blurry or off-domain results; VQ-GAN adds stroke effects but loses detail and symmetry. CLoSeR restores crisp contours and accurate strokes, resembling ground truth. For Sketches, base models distort proportions (e.g., bloated or muddy textures), whereas CLoSeR enhances contour sharpness and line stability. These improvements highlight its ability to recover fine-grained structure while embedding faithful stylistic cues.

### 4.2.3 USER STUDY

Table 1: User study preference rates.

| Datasets | CLoSeR | VQ-GAN | Base |
|---|---|---|---|
| MetFace | 70.8% | 16.9% | 12.3% |
| Monet | 83.3% | 12.5% | 4.2% |
| VanGogh | 71.7% | 13.3% | 15.0% |
| Ukiyo-e | 77.5% | 12.5% | 10.0% |
| *Average* | 75.8% | 13.8% | 10.4% |

We assess perceptual quality via user studies on both portrait (MetFace, 63 participants) and scene stylization (Monet, VanGogh, Ukiyo-e; 57 participants). In each trial, participants compare triplets from the *Base* model, *Vanilla VQ-GAN*, and *CLoSeR* and select the preferred result. As summarized in Tab. 1, CLoSeR is consistently favored across all datasets, indicating that the improvements in ArtFID/FID/KID align well with human judgments.

### 4.2.4 MODEL EFFICIENCY

As shown in Tab. 2, CLoSeR is highly efficient, requiring only 4.74 MB trainable parameters, 2.42 GB memory, and 0.0545 s inference—substantially lower than most baselines. Its lightweight test-time adaptation, without modifying the generator, offers an excellent trade-off between performance and resource cost, making it practical for low-resource applications.

### 4.3 MODEL ANALYSIS

**Ablation Study of CLoSeR.** Fig. 5 illustrates the progressive effect of our components on APDrawingGAN, AttenDistill, CAST, and StyleID. The base generators often produce blurry details and local artifacts; vanilla VQ-GAN sharpens textures but still exhibits geometric distortions, while adding positional encoding further improves spatial consistency. The full CLoSeR variant yields the sharpest

Table 2: Comparison of model efficiency.

| Methods | Params. (MB) | Memory (GB) | Time (s) |
|---|---|---|---|
| AdaAttN | 13.63 | 10.80 | 0.066 |
| CAST | 10.52 | 10.01 | 0.056 |
| AesPA-Net | 14.11 | 3.39 | 0.148 |
| StyleID | – | 12.87 | 5.848 |
| AttenDistill | 49.49 | 3.61 | 57.560 |
| **CLoSeR** | **4.74** | **2.42** | **0.055** |

geometry and cleanest textures, and Tab. 3 quantitatively confirms this trend, giving the best Art-FID/FID/KID across all four backbones.

**Ablation Study of codebook size.** We study the sensitivity of the codebook by just varying $K \in \{128, 256, 512, 1024\}$ on six backbones (AdaAttN, CAST, AesPA-Net, StyleID, AttenDistill, StyleSSP), reporting backbone-averaged ArtFID/FID/KID in Tab. 4. For all $K$, CLoSeR improves over the corresponding bases, indicating that it does not rely on a very large or carefully tuned code-

Table 3: Ablation study of CLoSeR.

| Method | APDrawingGAN ArtFID↓ | FID↓ | KID↓ | AttenDistill ArtFID↓ | FID↓ | KID↓ | CAST ArtFID↓ | FID↓ | KID↓ | StyleID ArtFID↓ | FID↓ | KID↓ |
|---|---|---|---|---|---|---|---|---|---|---|---|---|
| Base | 19.57 | 12.03 | 0.0267 | 28.16 | 18.74 | 0.0506 | 37.58 | 26.13 | 0.1002 | 35.60 | 23.78 | 0.1198 |
| + VQ-GAN | 19.54 | 11.96 | 0.0171 | 22.32 | 13.87 | 0.0271 | 37.22 | 25.37 | 0.1065 | 35.96 | 23.31 | 0.1080 |
| + VQ-GAN *w/* PE | 19.30 | 11.77 | 0.0170 | 22.24 | 13.74 | 0.0285 | 36.87 | 25.24 | 0.1057 | 35.44 | 22.98 | 0.1023 |
| + CLoSeR (Ours) | **18.70** | **11.35** | **0.0073** | **21.95** | **13.65** | **0.0255** | **36.11** | **24.50** | **0.0897** | **34.19** | **22.36** | **0.0966** |

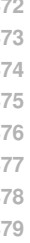
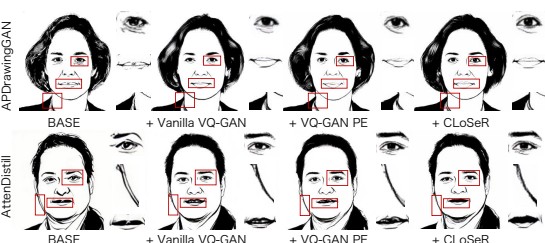
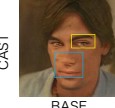
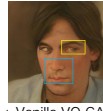
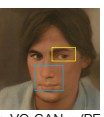
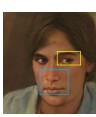
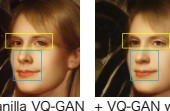
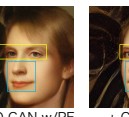

Figure 5: Qualitative ablation on APDrawing (*left*) and MetFace (*right*).

book. $K=1024$ achieves the best overall performance, with $K=256$ close behind, and the gaps across $K$ are modest, suggesting that CLoSeR is largely insensitive to the exact codebook size and that $K=1024$ is a reasonable default for continual style expansion.

**Validation of Continual Learning.** We assess continual learning by incrementally adding new tasks on both natural scene and portrait drawing datasets. Specifically, we adopt MetFace (Karras et al., 2020) as the style domain for faces (denoted as *Oil*), APDrawing (Yi et al., 2019) for pen drawings (*Pen*), and FS2K (Fan et al., 2022) for pencil sketches (*Pencil*), and Monet is used for natural scenes. As shown in Fig. 6, the refined models are evaluated on outputs from various base generators. The results demonstrate that performance on earlier styles remains largely stable even after introducing multiple new domains. These findings confirm that CLoSeR effectively mitigates catastrophic forgetting, retaining prior knowledge while adapting to new styles.

Table 4: Effect of codebook size $K$.

| Setting | MetFace ArtFID↓ | FID↓ | KID↓ | Monet ArtFID↓ | FID↓ | KID↓ |
|---|---|---|---|---|---|---|
| Base | 40.38 | 28.33 | 0.1262 | 25.87 | 16.19 | 0.0488 |
| + CLoSeR ($K=128$) | 36.29 | 24.42 | 0.0968 | 20.65 | 12.11 | 0.0354 |
| + CLoSeR ($K=256$) | 35.89 | 24.18 | 0.0966 | 20.41 | 11.96 | 0.0349 |
| + CLoSeR ($K=512$) | 36.20 | 24.45 | 0.0966 | 21.24 | 12.62 | 0.0360 |
| + CLoSeR ($K=1024$) | **34.91** | **24.13** | **0.0945** | **20.02** | **11.77** | **0.0337** |

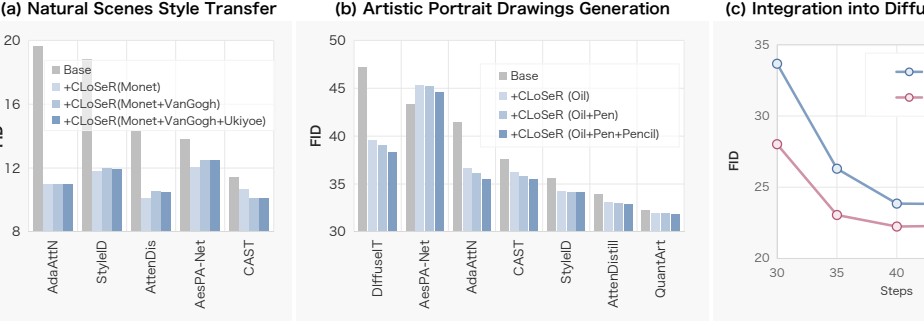

Figure 6: Catastrophic forgetting evaluation and integration into diffusion models. (a) Natural scenes style transfer with Monet as the target domain. (b) Artistic portrait drawings generation using Met-Face (Oil), APDrawing (Pen), and FS2K (Pencil). (c) Integration into StyleID (Chung et al., 2024) under varying sampling steps, where CLoSeR consistently reduces FID compared to the baseline.

Table 5: Validation of *Positional Encoding* (PE) with NME ($\downarrow$).

| Method | AdaAttN | AesPA-Net | CAST | DiffuseIT | StyleID | AttnDistill | Average |
|---|---|---|---|---|---|---|---|
| +VQ-GAN | 0.0357 | 0.0328 | 0.0348 | 0.0393 | **0.0338** | 0.0275 | 0.0340 |
| +VQ-GAN w/ PE | **0.0348** | **0.0314** | **0.0325** | **0.0377** | 0.0341 | **0.0274** | **0.0330** |

Table 6: Impact of CLoSeR on high-quality vs. degraded inputs.

| Setting | Method | AdaAttN | | | CAST | | | StyleID | | |
|---|---|---|---|---|---|---|---|---|---|---|
| | | ArtFID↓ | FID↓ | KID↓ | ArtFID↓ | FID↓ | KID↓ | ArtFID↓ | FID↓ | KID↓ |
| *High-quality* | Base | 33.13 | 21.94 | 0.0770 | 31.34 | 21.29 | 0.0564 | 29.42 | 19.52 | 0.0816 |
| | +CLoSeR | 30.30 | 19.90 | 0.0627 | 30.60 | 20.32 | 0.0550 | 28.44 | 18.54 | 0.0565 |
| | | ↓8.5% | ↓9.3% | ↓18.6% | ↓2.4% | ↓4.6% | ↓2.5% | ↓3.3% | ↓5.0% | ↓30.8% |
| *Degraded* | Base | 33.94 | 21.81 | 0.0752 | 31.22 | 20.16 | 0.0613 | 31.43 | 19.95 | 0.0786 |
| | +CLoSeR | 31.65 | 20.22 | 0.0736 | 30.82 | 19.74 | 0.0668 | 28.86 | 18.20 | 0.0576 |
| | | ↓6.7% | ↓7.3% | ↓2.1% | ↓1.3% | ↓2.1% | ↑9.0% | ↓8.2% | ↓8.8% | ↓26.7% |

**Validation of Positional Encoding.** To evaluate the role of positional encoding (PE) in geometric consistency, we adopt YOLOv5-face (Qi et al., 2022) as the evaluation backbone and test on stylized results from the MetFace dataset (Karras et al., 2020). We report *Normalized Mean Error* (NME) as the main metric. As shown in Tab. 5, PE consistently reduces NME across models, confirming its benefit in preserving geometric structure. Results with *Percentage of Correct Keypoints* (PCK) under different thresholds are provided in the appendix A.

**Integration into Diffusion Models.** We integrate CLoSeR into the StyleID (Chung et al., 2024) diffusion framework under varying sampling steps. As shown in Fig. 6(c), CLoSeR consistently reduces FID relative to the baseline, with improvements persisting across all iterations. This indicates that CLoSeR enhances domain alignment and stabilizes generation quality, even under fewer sampling steps. Additional qualitative results are provided in the appendix A.

**A Stress Test on Degraded Contents.** To assess the robustness of CLoSeR to input degradations, we conduct a stress test on corrupted contents. We randomly sample 20 style images from MetFace and 20 content images from CelebAMask-HQ, and apply a degradation pipeline to the contents that randomly combines Gaussian blur, multi-scale down–up sampling, Gaussian and Poisson noise, and JPEG compression (all at $256^2$ resolution). Using these degraded contents, we evaluate AdaAttN, CAST, and StyleID with and without CLoSeR under ArtFID/FID/KID (Tab. 6, Deg. means degraded). Even in this challenging setting, CLoSeR consistently improves over the baselines.

## 5 CONCLUSIONS AND LIMITATIONS

**Conclusions.** We presented CLoSeR, a lightweight test-time refinement framework that enhances style fidelity and geometric consistency for artistic style transfer. By combining LoRA-based continual adaptation, codebook expansion, and positional encoding, CLoSeR achieves parameter-efficient refinement while preserving prior knowledge across multiple domains. Extensive experiments on diverse benchmarks show consistent gains over GAN-, attention-, and diffusion-based baselines, together with strong robustness against catastrophic forgetting.

**Limitations and Future Work.** Despite these benefits, CLoSeR still inherits certain limitations from the underlying VQ-GAN backbone. When the input synthesis is severely distorted or lacks clear semantic structure, the refinement capacity becomes constrained. Moreover, our current design primarily targets spatial consistency, leaving finer temporal and semantic dynamics in video and multimodal settings underexplored. Extending CLoSeR to few-shot adaptation, video, and broader cross-modal applications is an important direction for future research.

## 6 ETHICS STATEMENT

This work does not involve human subjects, personally identifiable information, or sensitive data. All datasets used (e.g., MetFace, FS2K, APDrawing, Monet, VanGogh, Ukiyo-e) are publicly available and widely adopted in the literature. Our research focuses purely on artistic style transfer and does not raise foreseeable ethical or societal concerns such as bias, fairness, or privacy.

## 7 REPRODUCIBILITY STATEMENT

We have made every effort to ensure reproducibility. All model architectures, training strategies, and evaluation metrics (FID, KID, ArtFID, NME, PCK) are described in detail in the main paper and appendix. Additional implementation details, hyperparameters, and evaluation protocols are provided in the appendix A. We will release the source code upon publication to facilitate full reproducibility of our results.

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

# A APPENDIX

## A.1 IMPLEMENTATION DETAILS

In this section, we will provide a comprehensive overview of our experimental setup, detailing all aspects of the implementation to ensure transparency and reproducibility.

### A.1.1 CONTINUAL LEARNING BASED ON LORA

To enable scalable and memory-efficient continual learning in multi-style domains, we introduce Low-Rank Adaptation (LoRA) (Hu et al., 2022) into the VQ-GAN (Esser et al., 2021) framework. This is achieved by injecting LoRA modules into specific convolutional layers (conv1, conv2) of both the encoder and decoder. Each LoRA module performs a low-rank decomposition of the convolutional kernel updates, significantly reducing the number of trainable parameters during test-time refinement.

**Training Phase.** During training, the LoRA modules are initialized with a low-rank pair of trainable matrices $A \in \mathcal{R}^{r \times d_{in}}$, with a scaling factor $\alpha/r$. These modules are only activated for target style-specific adapters, each associated with a unique *style_id*. We implement a style-wise code isolation strategy by naming and registering all LoRA parameters under their respective *style_id*.In the continual learning scenario, only LoRA parameters and newly appended codebook embeddings are optimized, while all other original weights in the encoder, decoder, and quantizer are frozen. To accommodate novel style tokens without disrupting previously learned knowledge, we expand the codebook by appending new embeddings, and apply selective gradient masking to freeze the original indices. This ensures forward compatibility and avoids catastrophic forgetting.

**Inference Phase.**At test time, the framework dynamically selects and activates the appropriate LoRA module based on the input *style_id*. The inference pipeline searches for the latest LoRA checkpoint corresponding to the style domain, loads its parameters, and activates only the relevant LoRA paths while disabling others. This design ensures geometric consistency and stylistic specificity across diverse domains under a single model instance. Overall, the proposed LoRA-based continual adaptation mechanism provides a lightweight, modular, and effective solution to multi-style artistic synthesis, enabling test-time refinement with up to 94% fewer trainable parameters.

### A.1.2 DATASET DETAILS AND TRAINING CONFIGURATION

In this work, we employ three distinct datasets to train specialized codebooks for different artistic styles within our CLoSeR framework. Each dataset is carefully selected to represent a unique visual domain, enabling the learning of style-specific discrete representations.

**Artistic Portrait Generation.** We choose a pre-trained model (vqgan_metfaces_f16_1024.ckpt) from QuantArt (Huang et al., 2023) to finetune VQ-GAN (Esser et al., 2021) to achieve style-specific reconstruction. **MetFace** (Karras et al., 2020) is used to train the general facial appearance codebook. This dataset contains a total of 1336 face images, partitioned into 1,200 training samples and 136 test samples. **APDrawing** (Yi et al., 2019) datasets consist of pen-drawing portrait drawings. The dataset is divided into 420 training images and 70 test images. We initialize the VQ-GAN from the model pre-trained on the MetFace dataset (covering photorealistic facial appearances) and introduce Low-Rank Adaptation (LoRA) modules into the 'conv1' and 'conv2' of encoder and decoder. **FS2K** (Fan et al., 2022) includes 2,104 face sketches across three distinct artistic styles. We initialize the VQ-GAN from the model pre-trained on the APDrawing. We combine all three styles into a single training set to encourage the model to learn a more generalized sketch representation. The training split contains 2,004 images, with the remaining 100 reserved for testing.

**Scene Oil Paingting.** To further evaluate the generalization capability of our continual learning framework, we extend our experiments to three additional classical art styles:Monet,Van Gogh, and Ukiyo-e, all datasets are from WikiArt, follow the work from (Zhu et al., 2017). And we choose a pre-trained model (vqgan_wikiart_f16_1024.ckpt) from QuantArt (Huang et al., 2023) to finetune VQ-GAN to achieve style-specific scene oil painting reconstruction. **Monet** dataset comprises 1,072 training and 121 test images, capturing soft brushwork and natural light effects. **Van Gogh** dataset

Table 7: Impact of CLoSeR on *Natural Scenes Style Transfer*.

| Method | Monet | | | Vangogh | | | Ukiyo-e | | |
|---|---|---|---|---|---|---|---|---|---|
| | ArtFID ↓ | FID ↓ | KID ↓ | ArtFID ↓ | FID ↓ | KID ↓ | ArtFID ↓ | FID ↓ | KID ↓ |
| AdaAttN (CVPR'21) | 32.34 | 19.63 | 0.0602 | 23.60 | 13.55 | 0.0397 | 30.14 | 18.03 | 0.1380 |
| + VQGAN | 23.64 ↓26.9% | 13.86 ↓29.4% | 0.0681 ↓13.1% | 22.58 ↓4.3% | 12.88 ↓4.9% | 0.0259 ↓34.8% | 31.03 ↑3.0% | 18.36 ↑1.8% | 0.1144 ↓17.1% |
| + CLoSeR (ours) | 19.35 ↓40.2% | 10.99 ↓44.0% | 0.0467 ↓22.4% | 18.46 ↓21.8% | 10.35 ↓23.6% | 0.0277 ↓30.2% | 26.94 ↓10.6% | 15.77 ↓12.5% | 0.1193 ↓13.6% |
| CAST (SIGGRAPH'22) | 19.53 | 11.43 | 0.0159 | 24.90 | 14.81 | 0.0410 | 29.06 | 17.75 | 0.0888 |
| + VQGAN | 21.04 ↑7.7% | 12.11 ↑5.9% | 0.0240 ↑50.9% | 24.16 ↓3.0% | 14.00 ↓5.5% | 0.0297 ↓27.6% | 34.51 ↑18.8% | 20.63 ↑16.2% | 0.0710 ↓20.0% |
| + CLoSeR (ours) | 18.87 ↓3.3% | 10.64 ↓6.9% | 0.0155 ↓2.5% | 19.11 ↓23.3% | 10.89 ↓26.5% | 0.0250 ↓39.0% | 26.46 ↓8.9% | 15.47 ↓12.8% | 0.0814 ↓8.3% |
| AesPA-Net (ICCV'23) | 23.58 | 13.82 | 0.0808 | 22.7 | 12.99 | 0.0602 | 29.48 | 17.15 | 0.1673 |
| + VQGAN | 22.66 ↓3.9% | 12.98 ↓6.1% | 0.0631 ↓21.9% | 23.75 ↑4.6% | 13.64 ↑5.0% | 0.0309 ↓48.7% | 30.77 ↑4.4% | 17.85 ↑4.1% | 0.1479 ↓11.6% |
| + CLoSeR (ours) | 21.31 ↓9.6% | 12.02 ↓13.0% | 0.0639 ↓20.9% | 19.17 ↓15.6% | 10.81 ↓16.8% | 0.0330 ↓45.2% | 28.57 ↓3.1% | 16.45 ↓4.1% | 0.1458 ↓12.9% |
| StyleID (CVPR'24) | 23.81 | 15.07 | 0.0370 | 30.63 | 18.78 | 0.0532 | 32.39 | 20.04 | 0.1733 |
| + VQGAN (ours) | 22.94 ↓3.7% | 14.07 ↓6.6% | 0.0308 ↓16.8% | 28.74 ↓6.2% | 17.09 ↓9.0% | 0.0400 ↓24.8% | 32.95 ↑1.7% | 20.03 ↓0.0% | 0.1641 ↓5.3% |
| + CLoSeR (ours) | 19.85 ↓16.6% | 11.82 ↓21.6% | 0.0184 ↓50.1% | 22.11 ↓27.8% | 12.93 ↓31.2% | 0.0353 ↓33.6% | 27.13 ↓16.2% | 16.21 ↓19.1% | 0.1394 ↓19.6% |
| AtteneDist (CVPR'25) | 21.22 | 14.29 | 0.0489 | 22.13 | 14.24 | 0.0431 | 26.31 | 16.81 | 0.1718 |
| + VQGAN | 22.91 ↑8.0% | 14.91 ↑4.3% | 0.0320 ↓34.6% | 24.82 ↑12.2% | 15.54 ↑9.1% | 0.0353 ↓18.1% | 25.25 ↓4.0% | 15.7 ↓6.6% | 0.1388 ↓19.2% |
| + CLoSeR (ours) | 16.35 ↓23.0% | 10.1 ↓29.3% | 0.0216 ↓55.8% | 17.9 ↓19.1% | 10.82 ↓24.0% | 0.0245 ↓43.2% | 24.99 ↓5.0% | 14.34 ↓14.7% | 0.1259 ↓26.7% |
| StyleSSP (CVPR'25) | 34.72 | 22.85 | 0.0499 | 28.95 | 18.80 | 0.0628 | 28.39 | 18.30 | 0.1149 |
| + VQ-GAN | 30.60 ↓11.9% | 19.13 ↓16.3% | 0.0491 ↓1.6% | 26.34 ↓9.0% | 16.15 ↓14.1% | 0.0719 ↑14.5% | 26.96 ↓5.0% | 16.73 ↓8.6% | 0.1053 ↓8.4% |
| + CLoSeR (ours) | 24.41 ↓29.7% | 15.04 ↓34.2% | 0.0362 ↓27.5% | 22.28 ↓23.0% | 13.49 ↓28.2% | 0.0582 ↓7.3% | 24.63 ↓13.2% | 14.97 ↓18.2% | 0.1071 ↓6.8% |

includes 700 training and 100 test images, emphasizing expressive and vivid color contrasts. **Ukiyo-e** dataset contains 562 training and 263 test images, featuring flat color regions, strong outlines, and stylized compositions typical of traditional Japanese art.

All datasets are preprocessed to a consistent resolution of $256 \times 256$ with center cropping and normalized to the range $[-1, 1]$. During training, we preserve the LoRA parameters together with the corresponding discriminator for each style, enabling modular switching at inference time. This plug-and-play design supports flexible and memory-efficient multi-style generation within a single unified architecture.

### A.1.3  MORE METRICS DETAILS OF THE TASKS

We evaluate our model by ArtFID (Wright & Ommer, 2022), FID (Heusel et al., 2017), and KID (Bińkowski et al., 2018). The specific numerical metrics of Scene Oil Paintings are shown in the Table7, Face Portrait Drawings are shown in the Table 8. From the quantitative metrics, we can see that our algorithm has shown excellent performance under each base method.

### A.2  MORE RESULTS OF CLoSeR

**Image resolution.** All main experiments are conducted at an input resolution of $256^2$. To verify that this choice does not bias our conclusions, we additionally rerun the MetFace experiments on a single RTX 4090 at a higher resolution of $512^2$, keeping all hyper-parameters identical to the default setting except for reducing the batch size to 2. As summarized in Tab. 10, CLoSeR consistently improves ArtFID/FID/KID over the corresponding base generators for four representative backbones (AdaAttN, StyleID, AttenDistill, StyleSSP), while also producing visibly sharper and more faithful stylization. Since CLoSeR is a fully convolutional refinement module, it can in principle be applied to higher-resolution inputs without any architectural changes.

**Catastrophic Forgetting Evaluation of Continual Learning.** Due to space constraints, we report the detailed quantitative results of continual learning in the appendix. As shown in Table 11 and Table 12, the refined models are evaluated on outputs from various base generators. The results show that performance on earlier styles remains largely stable even after introducing multiple new domains. These findings confirm that **CLoSeR** effectively mitigates catastrophic forgetting, retaining prior knowledge while adapting to new styles.

**Validation of Positional Encoding.** To evaluate the role of positional encoding (PE) in geometric consistency, we conduct landmark detection on stylized outputs with CelebAMask-HQ (Lee et al.,

Table 8: Impact of CLoSeR on *Artistic Portrait Generation.*

| Method | Oil Painting | | | Pen Drawing | | | Sketch | | |
|---|---|---|---|---|---|---|---|---|---|
| | ArtFID ↓ | FID ↓ | KID ↓ | ArtFID ↓ | FID ↓ | KID ↓ | ArtFID ↓ | FID ↓ | KID ↓ |
| APDrawingGAN (CVPR'19) | – | – | – | 19.56 | 12.02 | 0.0267 | – | – | – |
| + VQGAN | – | – | – | 19.58 ↑0.1% | 11.96 ↓0.5% | 0.0171 ↓40.0% | – | – | – |
| + CLoSeR (ours) | – | – | – | 19.30 ↓1.3% | 11.77 ↓2.1% | 0.0073 ↓71.8% | – | – | – |
| AdaAttN (CVPR'21) | 41.39 | 28.14 | 0.1281 | 37.34 | 24.21 | 0.1293 | 44.81 | 26.74 | 0.0905 |
| + VQGAN | 35.26 ↓17.4% | 23.90 ↓17.7% | 0.1071 ↓19.6% | 29.57 ↓20.8% | 18.68 ↓22.8% | 0.0771 ↓40.4% | 43.87 ↓2.1% | 25.84 ↓3.4% | 0.0847 ↓6.4% |
| + CLoSeR (ours) | 36.62 ↓11.5% | 24.60 ↓12.9% | 0.1089 ↓14.9% | 29.11 ↓22.1% | 18.35 ↓24.2% | 0.0783 ↓39.5% | 41.04 ↓8.4% | 23.69 ↓11.4% | 0.1072 ↑18.4% |
| CAST (SIGGRAPH'22) | 37.58 | 26.13 | 0.1002 | 22.35 | 14.37 | 0.0784 | 43.07 | 25.41 | 0.0692 |
| + VQGAN | 37.22 ↓1.0% | 25.37 ↓2.9% | 0.1065 ↑6.3% | 23.70 ↑6.0% | 14.80 ↑3.0% | 0.0417 ↓46.8% | 40.61 ↓5.7% | 23.56 ↓7.3% | 0.0684 ↓1.2% |
| + CLoSeR (ours) | 36.11 ↓3.9% | 24.50 ↓6.2% | 0.1057 ↓5.5% | 23.55 ↑5.4% | 14.68 ↑2.2% | 0.0417 ↓46.8% | 40.28 ↓6.5% | 22.86 ↓10.0% | 0.0841 ↑21.5% |
| AesPA-Net (ICCV'23) | 43.28 | 30.42 | 0.1313 | 41.97 | 28.53 | 0.1258 | 41.52 | 25.11 | 0.0955 |
| + VQGAN | 44.25 ↑2.2% | 30.48 ↑0.2% | 0.1450 ↑10.4% | 26.59 ↓36.6% | 16.74 ↓41.3% | 0.0464 ↓63.1% | 40.50 ↓2.5% | 23.78 ↓5.3% | 0.0629 ↓34.1% |
| + CLoSeR (ours) | 45.05 ↑4.1% | 31.07 ↑2.1% | 0.1314 ↓0.0% | 27.35 ↓34.9% | 17.23 ↓39.6% | 0.0497 ↓60.5% | 38.49 ↓7.3% | 21.97 ↓12.5% | 0.0727 ↓23.9% |
| DiffuseIT (ICLR'23) | 47.13 | 32.27 | 0.1598 | 36.19 | 23.06 | 0.0826 | 58.04 | 35.86 | 0.1858 |
| + VQGAN | 48.91 ↑3.8% | 32.70 ↑1.3% | 0.1110 ↓30.5% | 30.57 ↓18.4% | 18.27 ↓26.2% | 0.0646 ↓27.9% | 46.33 ↓20.2% | 26.91 ↓25.0% | 0.0739 ↓60.2% |
| + CLoSeR (ours) | 39.38 ↓16.4% | 26.46 ↓17.7% | 0.0913 ↓42.9% | 32.06 ↓11.4% | 19.24 ↓16.6% | 0.0557 ↓32.6% | 41.86 ↓27.9% | 23.70 ↓33.9% | 0.0807 ↓56.6% |
| InST (CVPR'23) | 57.89 | 38.57 | 0.2226 | 35.04 | 26.13 | 0.0818 | – | – | – |
| + VQGAN | 46.46 ↓19.7% | 32.11 ↓16.7% | 0.0957 ↓57.0% | 31.24 ↓10.8% | 20.01 ↓23.4% | 0.0783 ↓4.3% | – | – | – |
| + CLoSeR (ours) | 47.23 ↓18.4% | 26.46 ↓31.4% | 0.0913 ↓58.7% | 29.72 ↓15.2% | 19.08 ↓27.0% | 0.0779 ↓4.8% | – | – | – |
| StyleID (CVPR'24) | 35.60 | 23.78 | 0.1198 | 26.58 | 17.44 | 0.0235 | 44.67 | 26.97 | 0.1546 |
| + VQGAN | 35.96 ↑1.0% | 23.31 ↓2.0% | 0.1080 ↓9.8% | 22.05 ↓17.0% | 13.71 ↓21.4% | 0.0235 ↓0.0% | 41.01 ↓8.2% | 23.96 ↓11.2% | 0.0643 ↓58.4% |
| + CLoSeR (ours) | 34.19 ↓4.0% | 22.36 ↓6.0% | 0.0966 ↓19.4% | 24.57 ↓7.6% | 15.43 ↓11.5% | 0.0159 ↓32.3% | 36.14 ↓19.1% | 20.87 ↓22.6% | 0.0725 ↓53.1% |
| AttenDist (CVPR'25) | 33.95 | 26.13 | 0.1349 | 28.16 | 18.74 | 0.0506 | 43.50 | 30.98 | 0.1501 |
| + VQGAN | 34.26 ↑0.9% | 24.66 ↓5.6% | 0.1162 ↓13.9% | 22.32 ↓20.7% | 13.87 ↓26.0% | 0.0271 ↓46.4% | 46.47 ↑6.8% | 28.43 ↓8.2% | 0.0798 ↓46.8% |
| + CLoSeR (ours) | 33.18 ↓2.3% | 23.84 ↓8.8% | 0.1046 ↓22.5% | 21.95 ↓22.1% | 13.65 ↓27.2% | 0.0255 ↓49.6% | 39.59 ↓9.0% | 23.89 ↓22.9% | 0.0843 ↓43.8% |
| StyleSSP (CVPR'25) | 46.59 | 33.49 | 0.1143 | 29.57 | 20.24 | 0.1119 | 40.44 | 26.07 | 0.1320 |
| + VQ-GAN | 43.66 ↓6.3% | 31.16 ↓7.0% | 0.0872 ↓23.7% | 28.49 ↓3.7% | 18.15 ↓10.3% | 0.0691 ↓38.2% | 44.25 ↑9.4% | 26.31 ↑0.9% | 0.0708 ↓46.4% |
| + CLoSeR (ours) | 40.25 ↓13.6% | 28.14 ↓16.0% | 0.0750 ↓34.4% | 28.07 ↓5.1% | 17.85 ↓11.8% | 0.0644 ↓42.4% | 37.74 ↓6.7% | 22.43 ↓14.0% | 0.0833 ↓36.9% |

Table 9: Validation of Positional Encoding (PE) with PCK (↑).

| Metrics | AdaAttN | | AesPA-Net | | AttnDistill | | CAST | | DiffuseIT | |
|---|---|---|---|---|---|---|---|---|---|---|
| | +VQ-GAN | +VQ-GAN w/PE | +VQ-GAN | +VQ-GAN w/PE | +VQ-GAN | +VQ-GAN w/PE | +VQ-GAN | +VQ-GAN w/PE | +VQ-GAN | +VQ-GAN w/PE |
| PCK@5% ↑ | 0.789 | **0.802** | 0.841 | **0.858** | 0.896 | **0.901** | 0.806 | **0.842** | 0.741 | **0.764** |
| PCK@7% ↑ | 0.921 | **0.930** | 0.942 | **0.952** | 0.973 | **0.973** | 0.927 | **0.941** | 0.900 | **0.909** |
| PCK@10% ↑ | 0.980 | **0.987** | 0.984 | **0.988** | 0.995 | **0.995** | 0.979 | **0.986** | 0.974 | **0.980** |

2020) as the content domain and MetFace (Karras et al., 2020) as the style domain. For each algorithm, we generate 80 stylized results, where both the vanilla VQ-GAN and VQ-GAN w/PE are trained on MetFace for 48 epochs. The qualitative comparisons of different detection algorithms are provided in Figure 8. Due to space constraints, additional *Percentage of Correct Keypoints* (PCK) results under 5%, 7%, and 10% thresholds are reported in the Appendix, as shown in Table 9.

**Integration into Diffusion Models.** We integrate CLoSeR into the StyleID (Chung et al., 2024) diffusion framework and evaluate under different sampling steps. As shown in Figure 7, we assess refinement on MetFace-based generations at 30, 35, 40, and 50 steps. The qualitative results clearly demonstrate that CLoSeR produces sharper and more stylistically faithful portraits across different iteration counts.

**Multi-round refinement.** We further study whether applying CLoSeR multiple times brings additional benefits by running one (1×) or two (2×) refinement passes on MetFace for three representative backbones (Tab. 13). While the first pass yields clear gains over the Base and +VQ-GAN variants, the differences between 1× and 2× refinement are negligible (changes ≤ 0.1 ArtFID and ≤ 0.2 FID in all cases). This suggests that the initial pass already projects the latent features close to their optimal codebook representations, so an extra pass produces almost identical reconstructions; CLoSeR effectively behaves as an approximately idempotent projection onto the learned VQ manifold. Therefore, we adopt a single refinement round in all main experiments, as additional rounds only increase inference time without measurable quality gains.

Table 10: Effect of increasing the input resolution to $512 \times 512$ on MetFace.

| Method | Row | ArtFID ↓ | FID ↓ | KID ↓ |
|---|---|---|---|---|
| AdaAttN (CVPR'21)) | Base | 31.54 | 21.06 | 0.0696 |
| | + CLoSeR | 30.19 | 20.33 | 0.0728 |
| | | 4.3% ↓ | 3.5% ↓ | 4.6% ↑ |
| StyleID (CVPR'24)) | Base | 30.56 | 20.97 | 0.0588 |
| | + CLoSeR | 28.23 | 19.38 | 0.0547 |
| | | 7.6% ↓ | 7.6% ↓ | 7.0% ↓ |
| AttenDistill (CVPR'25)) | Base | 31.01 | 22.40 | 0.0650 |
| | + CLoSeR | 27.26 | 19.37 | 0.0589 |
| | | 12.1% ↓ | 13.5% ↓ | 9.4% ↓ |
| StyleSSP (CVPR'25)) | Base | 38.76 | 25.49 | 0.0830 |
| | + CLoSeR | 31.39 | 21.24 | 0.0722 |
| | | 19.0% ↓ | 16.7% ↓ | 13.0% ↓ |

Table 11: Catastrophic Forgetting Evaluation on artistic portrait (MetFace, APDrawing, FS2K).

| Methods | ArtFID ↓ | FID ↓ | KID ↓ |
|---|---|---|---|
| AdaAttN | 41.40 | 28.15 | 0.1281 |
| +CLoSeR (Oil) | 36.61 | 24.60 | 0.1089 |
| +CLoSeR (Oil+Pen) | 36.11 | 24.24 | **0.1067** |
| +CLoSeR (Oil+Pen+Pencil) | **35.44** | **23.69** | 0.1106 |
| CAST | 37.58 | 26.13 | **0.1002** |
| +CLoSeR (Oil) | 36.22 | 24.52 | 0.1055 |
| +CLoSeR (Oil+Pen) | 35.79 | 24.21 | 0.1045 |
| +CLoSeR (Oil+Pen+Pencil) | **35.53** | **24.03** | 0.1093 |
| AesPA-Net | **43.28** | **30.42** | 0.1313 |
| +CLoSeR (Oil) | 45.34 | 31.25 | **0.1304** |
| +CLoSeR (Oil+Pen) | 45.17 | 31.07 | 0.1321 |
| +CLoSeR (Oil+Pen+Pencil) | 44.61 | 30.60 | 0.1375 |
| QuantArt | 32.29 | 24.43 | 0.1017 |
| +CLoSeR (Oil) | 31.98 | 23.59 | **0.0917** |
| +CLoSeR (Oil+Pen) | 31.89 | 23.47 | 0.0929 |
| +CLoSeR (Oil+Pen+Pencil) | **31.83** | **23.41** | 0.0958 |
| DiffuseIT | 47.13 | 32.27 | 0.1598 |
| +CLoSeR (Oil) | 39.59 | 26.60 | **0.0913** |
| +CLoSeR (Oil+Pen) | 39.07 | 26.24 | 0.0919 |
| +CLoSeR (Oil+Pen+Pencil) | **38.30** | **25.65** | 0.0929 |
| StyleID | 35.60 | 23.78 | 0.1198 |
| +CLoSeR (Oil) | 34.19 | 22.36 | **0.0966** |
| +CLoSeR (Oil+Pen) | **34.14** | **22.31** | 0.0971 |
| +CLoSeR (Oil+Pen+Pencil) | **34.14** | **22.31** | 0.0971 |
| AttenDistill | 33.95 | 26.13 | 0.1349 |
| +CLoSeR (Oil) | 33.04 | 23.71 | **0.1041** |
| +CLoSeR (Oil+Pen) | 33.00 | 23.61 | 0.1043 |
| +CLoSeR (Oil+Pen+Pencil) | **32.87** | **23.50** | 0.1089 |

### A.2.1 EXPERIMENTS RESULTS

In this section, we provide a concise yet comprehensive overview of the additional experimental results validating our proposed Continual Learning for Style Refinement (CLoSeR) framework. We compare CLoSeR with state-of-the-art methods, emphasizing its effectiveness in generating high-quality, style-consistent drawings.

**Facial Portrait Results.** Figure 9 and Figure 10 illustrate the generated artistic styles for facial portraits using various generative methods. CLoSeR demonstrates superior performance in both oil painting and pen drawing styles. For oil paintings, CLoSeR achieves visually appealing results that

Table 12: Catastrophic Forgetting Evaluation on natural scenes (Monet, Van Gogh, Ukiyo-e).

| Methods | ArtFID ↓ | FID ↓ | KID ↓ |
|---|---|---|---|
| AdaAttN | 32.34 | 19.63 | 0.0602 |
| +CLoSeR (+Monet) | 19.35 | 10.99 | **0.0467** |
| +CLoSeR (+Monet+VanGogh) | **19.27** | **10.95** | 0.0513 |
| +CLoSeR (+Monet+VanGogh+Ukiyo-e) | **19.27** | **10.95** | 0.0493 |
| CAST | 19.53 | 11.43 | 0.0159 |
| +CLoSeR (+Monet) | 18.87 | 10.64 | 0.0155 |
| +CLoSeR (+Monet+VanGogh) | **18.03** | **10.11** | **0.0117** |
| +CLoSeR (+Monet+VanGogh+Ukiyo-e) | 18.05 | 10.13 | 0.0118 |
| AesPA-Net | 23.58 | 13.82 | 0.0808 |
| +CLoSeR (+Monet) | **21.31** | **12.02** | **0.0639** |
| +CLoSeR (+Monet+VanGogh) | 22.05 | 12.48 | 0.0677 |
| +CLoSeR (+Monet+VanGogh+Ukiyo-e) | 22.05 | 12.48 | 0.0677 |
| StyleID | 30.63 | 18.78 | 0.0370 |
| +CLoSeR (+Monet) | **19.85** | **11.82** | 0.0184 |
| +CLoSeR (+Monet+VanGogh) | 20.07 | 11.98 | 0.0165 |
| +CLoSeR (+Monet+VanGogh+Ukiyo-e) | 19.96 | 11.91 | **0.0159** |
| AttenDistill | 21.22 | 14.29 | 0.0489 |
| +CLoSeR (+Monet) | **16.35** | **10.10** | **0.0216** |
| +CLoSeR (+Monet+VanGogh) | 17.03 | 10.54 | 0.0267 |
| +CLoSeR (+Monet+VanGogh+Ukiyo-e) | 16.97 | 10.50 | 0.0257 |

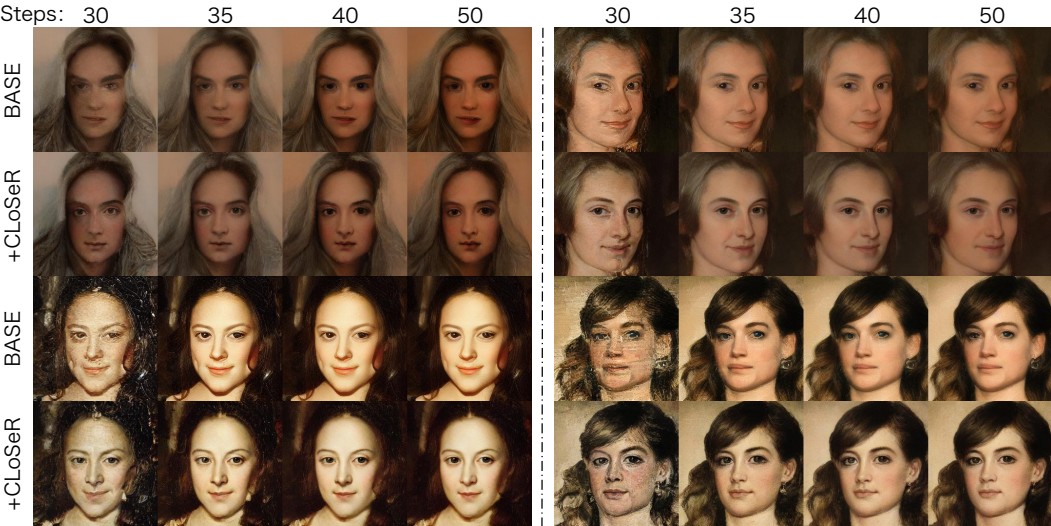

Figure 7: Qualitative evaluation of StyleID refinement with CLoSeR on the MetFace dataset under different sampling steps (30, 35, 40, 50). Compared to the baseline (BASE), CLoSeR produces sharper, more consistent, and stylistically faithful results across all iterations.Please zoom in for details.

closely resemble the target style while preserving the identity and structural details of the input faces. Compared to the SOTA methods, CLoSeR avoids overly smoothed or distorted outputs, capturing complex brush strokes and color blending effectively. In pen drawings, CLoSeR produces clear lines and consistent textures, accurately representing the input faces with sharp, well-defined lines.

**Natural Scene Results**   Figure 11 showcases the generated artistic styles for natural scenes based on different artist and generative methods. CLoSeR excels in generating high-quality artistic representations of natural scenes, such as Monet, Van Gogh, and Ukiyo-e styles. For Monet's impressionistic style, CLoSeR captures soft brushwork and natural light effects, producing visually

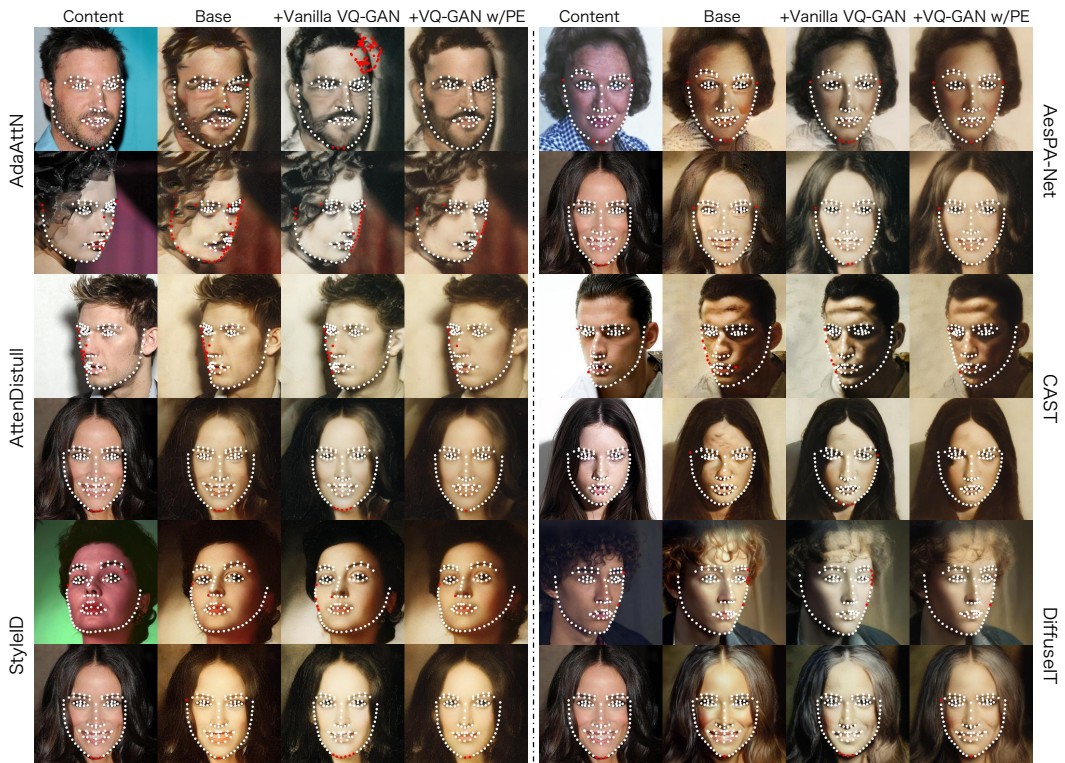

Figure 8: Comparison of facial landmark detection across different generative methods on oil painting portraits. Each column shows the detected landmarks on stylized outputs, highlighting the impact of VQ-GAN and positional encoding (PE) on geometric consistency. Please zoom in for details.

Table 13: Effect of single– vs. double–round refinement on MetFace (ArtFID↓/FID↓/KID↓).

| Method | AdaAttN | | | DiffuseIT | | | StyleID | | |
|---|---|---|---|---|---|---|---|---|---|
| | ArtFID | FID | KID | ArtFID | FID | KID | ArtFID | FID | KID |
| Base | 41.39 | 28.14 | 0.1281 | 47.13 | 32.27 | 0.1598 | 35.60 | 23.78 | 0.1198 |
| + VQGAN | 35.26 | 23.90 | 0.1071 | 48.91 | 32.70 | 0.1110 | 35.96 | 23.31 | 0.1080 |
| + CLoSeR (1×) | 36.62 | 24.60 | 0.1089 | 39.38 | 26.46 | 0.0913 | 34.19 | 22.36 | 0.0966 |
| + CLoSeR (2×) | 36.61 | 24.60 | 0.1088 | 39.59 | 26.60 | 0.0912 | 34.19 | 22.36 | 0.0966 |

pleasing results. In Van Gogh's post-impressionistic style, CLoSeR effectively reproduces expressive, swirling strokes and vivid color contrasts. For Ukiyo-e, CLoSeR generates flat color regions, strong outlines, and stylized compositions typical of traditional Japanese art. Compared to other SOTA methods, CLoSeR maintains better style consistency and visual fidelity. The continual learning approach ensures that CLoSeR refines its understanding of each artistic style, leading to more accurate and consistent results.

## A.3 USAGE OF LLM

We employed a large language model (LLM) as an auxiliary tool during the manuscript preparation process. Specifically, the LLM was used to polish the writing, check spelling and grammar errors, and improve the overall clarity and readability of the text. Importantly, the LLM was not involved in designing the methodology, conducting experiments, or analyzing results; all technical contributions, experimental designs, and conclusions were developed solely by the authors. The use of the LLM was limited to language refinement, helping to ensure that the presentation of our work is logically coherent and accessible to a broader research audience.

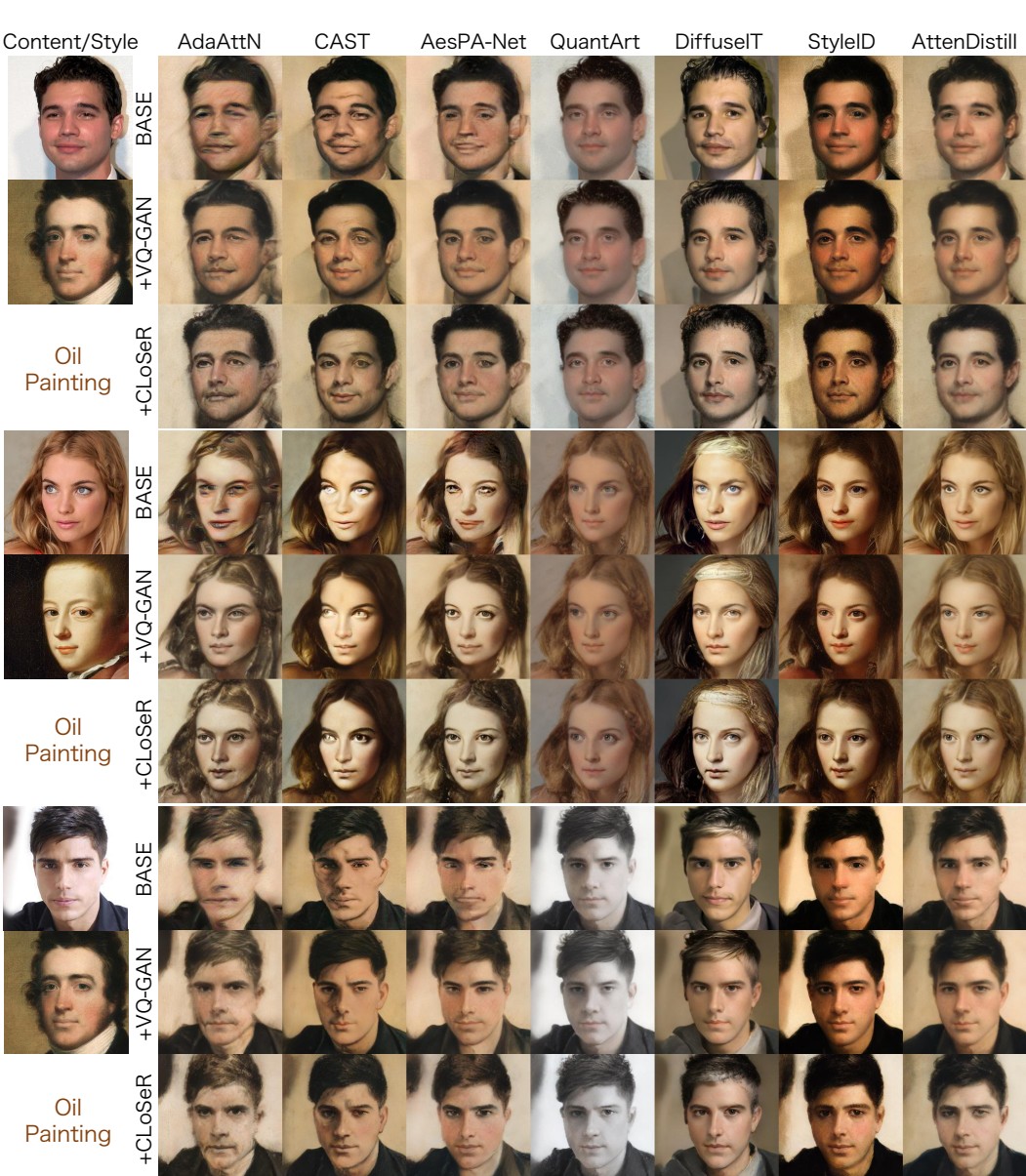

Figure 9: Generated results of artistic styles for oil facial portraits based on different generative methods.Please zoom in for details.

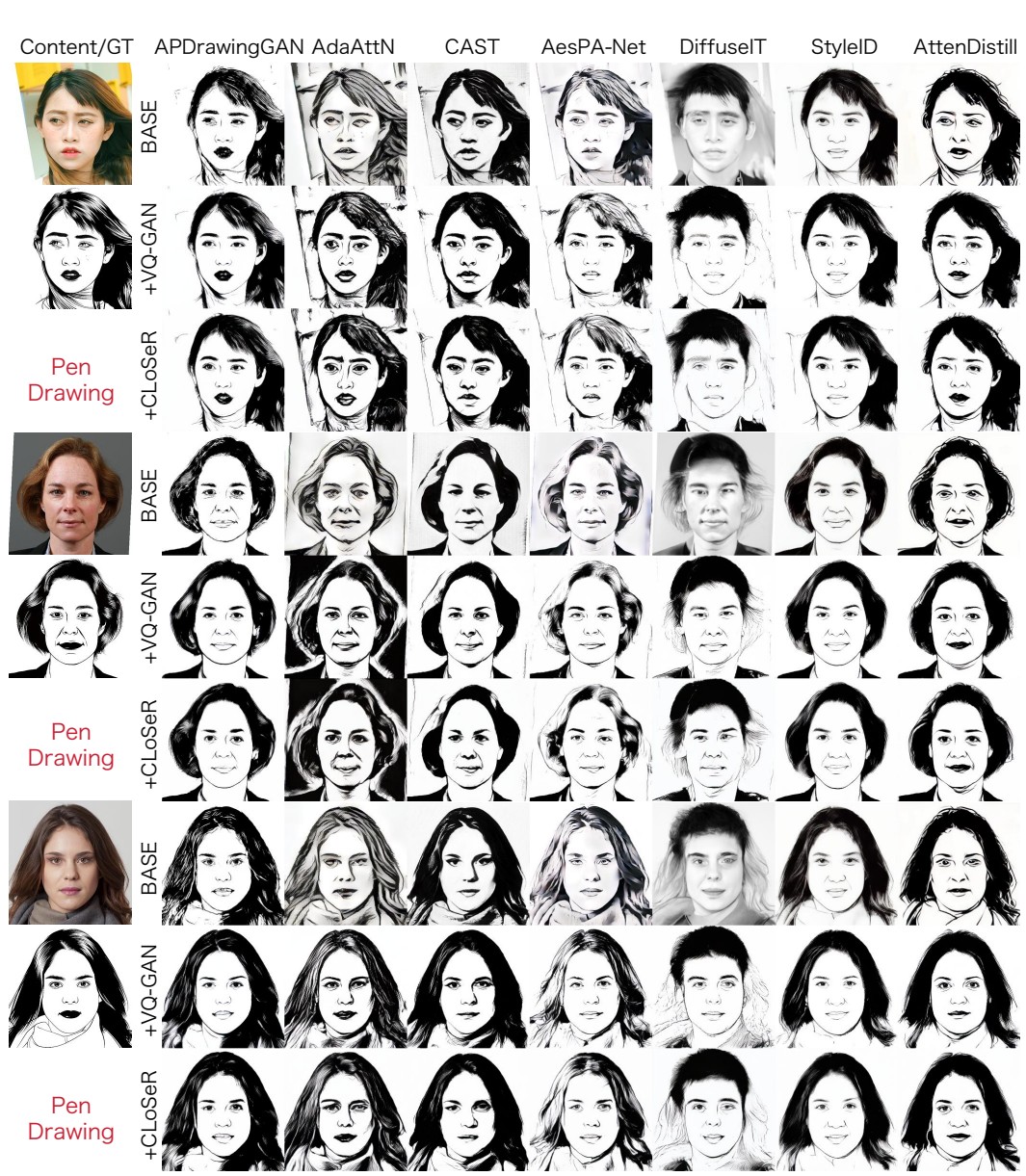

Figure 10: Generated results of artistic styles for pen-drawing facial portraits based on different generative methods.Please zoom in for details.

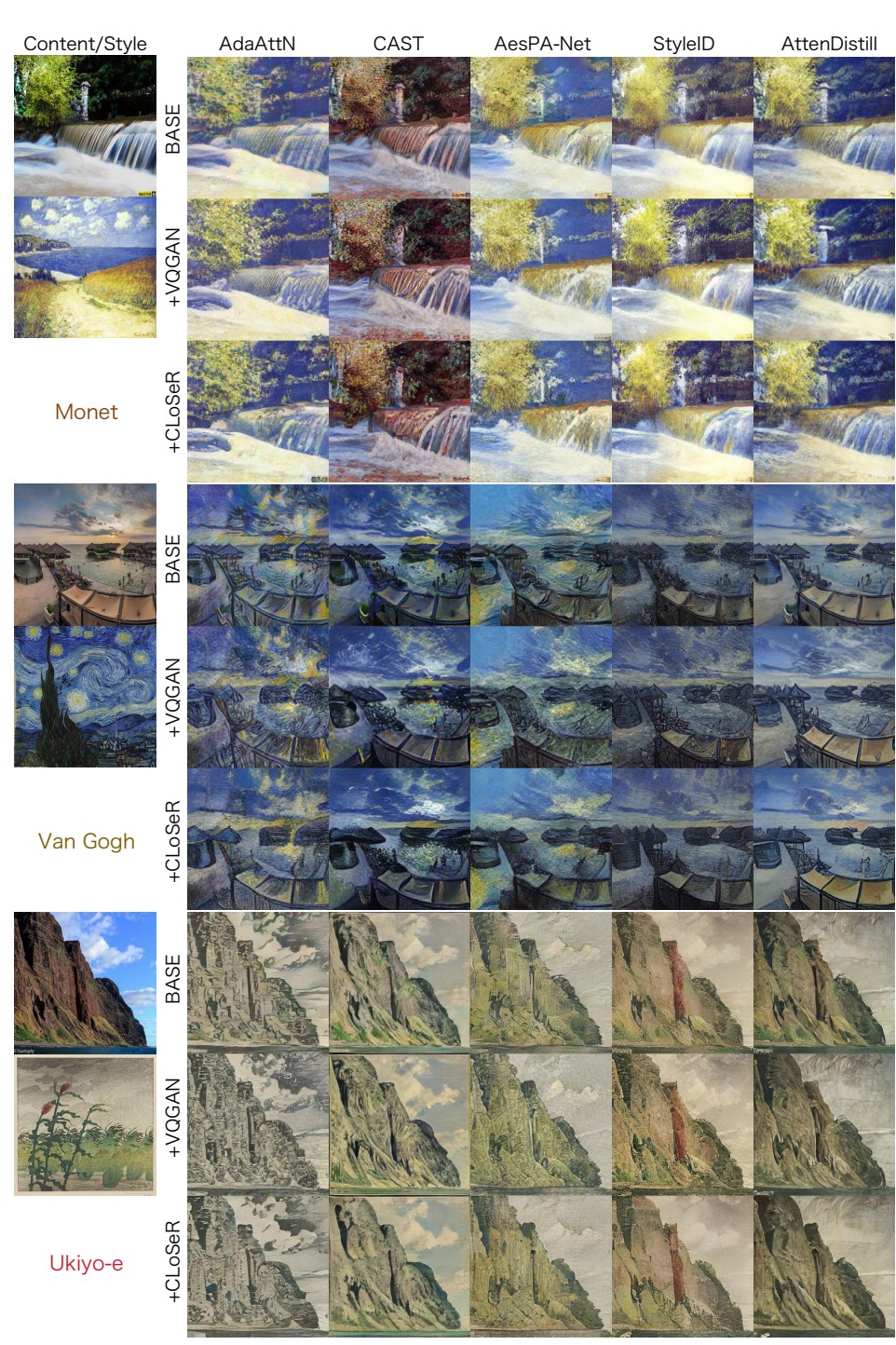

Figure 11: Generated results of artistic styles for natural scenes based on different artist and generative methods.Please zoom in for details.

