# OpenReview forum: "CLoSeR: Continual Learning in VQ-GAN for Test-time Style Refinement"
_ICLR.cc/2026/Conference — Submitted to ICLR 2026_

### Official Review · Reviewer_a6PH · 2025-10-26

**Soundness:** 3
**Presentation:** 3
**Contribution:** 2
**Rating:** 6
**Confidence:** 2

**Summary:**

This paper proposes an universal test-time-refinement (TTR) framework for improving generation quality and style alignment of image artistic style transfer (AST). Specifically, this method employs a self-supervised VQ-GAN model as a post-processing module which is trained to reconstruct the input, effectively enhancing the outputs from any AST generator. To adapt to a new artistic style, this VQ-GAN model can be trained at test-time through continual learning via LoRA and incremental codebook expansion. To enhance the geometric prior of the model, the quantization in VQ-GAN is augmented with 2D positional embeddings. Together, the proposed method CLoSeR can be used as an universal post-processing model for any AST models including neural style-transfer and diffusion-based stylized image generation.

**Strengths:**

- The proposed method is efficient and is orthogonal to other AST model designs. As presented in Section 4.2.3 and Table 1, CLoSeR contains only 4.74 MB trainable parameters. Notably, it takes 0.055 seconds for inference, which is negligible compared to those generators with diffusion models.
- The effectiveness of CLoSeR is demonstrated through extensive experiments on different AST generators including neural style-transfer and diffusion-based stylized image generation. CLoSeR shows consistent improvements on multiple quantitative metrics.
- The proposed framework is robust against novel artistic styles outside the VQ-GAN training data. The VQ-GAN can be trained on the new style data in a self-supervised manner. Moreover, this continual learning process is implemented through LoRA and incremental codebook updates, enhancing the efficiency of the proposed method.
- The ablation experiments’ configuration is reasonable.

**Weaknesses:**

- Despite the constant improvements of CLoSeR on different generators, its performances rely on the baseline. As presented in Figure 3, CLoSeR on more powerful base generators like AttenDis has better quantitative metrics.
- The connection between a VQ-GAN refiner (for improving style fidelity) and continual learning framework is somehow weak. The improved geometric consistency in CLoSeR is justified through ablation, while the benefit of continual learning in this task is unclear. According to my understanding, the refiner is simply trained to ‘overfit’ to one artistic style at a time.
- There are some typos in equation 2.

**Questions:**

- The quantization in VQ-GAN is known to introduce some discretization error in reconstruction, potentially suffering the risk of reduced variety in the generation outputs. How could the proposed method handle the reconstruction error in VQ-GAN?
- Is it possible to let the CLoSeR refine the outputs for multi rounds? Like using the first refinement output as the input to the second round and so on.

---

> ### Author Response · Authors · 2025-12-01
> **Response to Reviewer a6PH**
>
> We thank the reviewer for the positive assessment of our efficiency, robustness, and experimental configuration, and for the constructive questions. We address the concerns below.
>
> **(1) Performance relies on the baseline; AttenDis has better quantitative metrics.**
>
> We would like to clarify that CLoSeR is designed as `a universal test-time refinement module` rather than a new standalone AST backbone. Thus, the appropriate comparison is between each generator and its refined counterpart. Across all seven generators, including the strong AttenDistill baseline, attaching CLoSeR `consistently improves ArtFID, FID, and KID`. For weaker backbones, these gains often `close the gap to or even surpass stronger baselines`, while AttenDistill itself still benefits from CLoSeR. We have emphasized in the revision that CLoSeR is `orthogonal to backbone design`: stronger generators naturally yield stronger refined models, and our contribution is a lightweight plug-in that reliably boosts existing AST systems without modifying or retraining the generators.
>
>
> **(2) Connection between VQ-GAN refiner and continual learning is weak; seems to overfit one style.**
>
> We clarify that continual learning is a core design goal of CLoSeR. It is instantiated by **(i) incremental codebook expansion**, where each new style only allocates a small set of additional codes while previous styles continue to use their original entries, and **(ii) sequential LoRA-based adaptation**, where each new style is learned by inserting a small set of low-rank adapters on top of the previously trained refiner instead of retraining the whole model from scratch. In this way, a single CLoSeR instance can be continually extended to handle multiple artistic styles within one unified model. The LoRA modules account for only about 6% of the parameters, `reducing the number of trainable weights by roughly 94% compared with full fine-tuning`; in practice we only store the compact LoRA checkpoints and dynamically load them at inference time, which substantially saves disk storage. As shown in `Fig. 6` and `Tab.7--8 (appendix)`, after extending from Monet$\rightarrow$VanGogh$\rightarrow$Ukiyo-e and MetFace$\rightarrow$APDrawing$\rightarrow$FS2K, CLoSeR `preserves or even slightly improves performance on earlier styles`, indicating `parameter-efficient continual adaptation` rather than overfitting to the last style.
>
> **(3) Typos in equation (2).**
>
> We thank the reviewer for spotting these issues. We have corrected the typos in Eq.(2) and carefully re-check all mathematical expressions in the revised version.
>
>
> **(4) Quantization error and possible loss of diversity.**
>
> We acknowledge that vector quantization may introduce discretization error. In CLoSeR, this risk is mitigated by (i) a strong reconstruction objective (L1 + LPIPS + adversarial loss) that encourages the VQ-GAN codebook to preserve both structure and texture, and (ii) incremental codebook expansion, which increases representational capacity as new styles are added.
>
> Empirically, we do not observe `any sign of diversity collapse`. On the contrary, **KID—which is particularly sensitive to mode collapse—consistently decreases** after applying CLoSeR. As reported in `Tab.4, Tab.5, Tab.7 and Tab.8 (in the appendix)`, CLoSeR improves KID `across all generators` (AdaAttN, CAST, AesPA-Net, DiffuseIT, StyleID, AttenDistill, StyleSSP, QuantArt) and across `both natural-scene and portrait domains`, often with relative reductions of 10–40\%. These systematic KID gains, together with the concurrent improvements in ArtFID/FID and the richer visual details in our qualitative results, indicate that CLoSeR not only maintains but in fact enhances sample diversity while improving stylistic fidelity.
>
> Moreover, when CLoSeR is continually extended to new tasks, the encoder–decoder and codebook are exposed to a broader distribution of stylized inputs, effectively increasing the diversity of learned codes and acting as a regularizer against overfitting to a single style. This effect is reflected in our catastrophic-forgetting evaluation: as shown in `Tab.7 and Tab.8 (in the appendix)`, adding new styles (e.g., Oil $\rightarrow$ Oil+Pen $\rightarrow$ Oil+Pen+Pencil, or Monet $\rightarrow$ Monet+VanGogh $\rightarrow$ Monet+VanGogh+Ukiyo-e) `not only preserves but often slightly improves ArtFID/FID/KID on earlier domains`. Together, these observations suggest that **CLoSeR does not harm, and can even benefit, diversity and generalization**.

---

> ### Author Response · Authors · 2025-12-01
> **Response to Reviewer a6PH**
>
> **(5) On multi–round refinement.**
>
> To examine whether repeated refinement brings additional benefits, we apply CLoSeR once (1$\times$) or twice (2$\times$) on the MetFace domain and report results for three representative backbones in the table below:
>
> | Method                    | AdaAttN |           |           | DiffuseIT |           |      | StyleID |      |           |
> |---------------------------|-----------|---------|-----------|-----------|-----------|-----------|------|------|------|
> |                           | ArtFID ↓  | FID ↓   | KID ↓     | ArtFID ↓  | FID ↓     | KID ↓     | ArtFID ↓ | FID ↓ | KID ↓ |
> | Base              | 41.39            | 28.14  | 0.1281  | 47.13               | 32.27  | 0.1598  | 35.60             | 23.78  | 0.1198  |
> | + VQGAN           | 35.26            | 23.90  | 0.1071  | 48.91               | 32.70  | 0.1110  | 35.96             | 23.31  | 0.1080  |
> | + CLoSeR (1×)     | 36.62            | 24.60  | 0.1089  | 39.38               | 26.46  | 0.0913  | 34.19             | 22.36  | 0.0966  |
> | + CLoSeR (2×)     | 36.61            | 24.60  | 0.1088  | 39.59               | 26.60  | 0.0912  | 34.19             | 22.36  | 0.0966  |
>
> Compared with the gains obtained by attaching CLoSeR to the base generators, the difference between 1$\times$ and 2$\times$ refinement is negligible (changes $<0.1$ ArtFID and $<0.2$ FID in all cases). This suggests that, after the first pass, `the latent features have already been projected close to their optimal codebook representations`, so an additional pass produces almost identical reconstructions. In other words, CLoSeR behaves approximately as an idempotent projection onto the learned VQ manifold. Therefore, we adopt a single refinement round in the main experiments, as `extra rounds increase inference time without providing measurable quality gains`.

---

### Official Review · Reviewer_pDDG · 2025-10-28

**Soundness:** 2
**Presentation:** 3
**Contribution:** 2
**Rating:** 4
**Confidence:** 4

**Summary:**

This paper proposes a lightweight test-time refinement framework, called CLoSeR, that improves style fidelity and geometric consistency in artistic style transfer. CLoSeR consists of three key components: LoRA-based continual adaptation that injects trainable low-rank matrices to modulate features in a style-specific manner, codebook expansion that enables the model to encode style-specific visual primitives while preserving previously learned representations, and positional encoding that enables geometry-aware refinement without introducing any additional learnable parameters.

**Strengths:**

1. The proposed method can be applied to current style transfer approaches to enhance the quality of the generated images.
2. This paper is well-written and well-organized.
3. Extensive experiments are conducted to evaluate the performance of the proposed method.

**Weaknesses:**

1. As shown in Figure 4, while the method proposed in this paper offers some improvement over the original baseline, the difference is not significant, and the extent of the improvement is relatively modest and not very satisfactory.

2. Many state-of-the-art style transfer methods are not introduced and compared in this paper, such as SaMam [1], HIS [2], OmniStyle [3], and StyleSSP [4]. \
[1] SaMam: Style-aware State Space Model for Arbitrary Image Style Transfer. CVPR 2025. \
[2] HSI: A Holistic Style Injector for Arbitrary Style Transfer. CVPR 2025. \
[3] OmniStyle: Filtering High Quality Style Transfer Data at Scale. CVPR 2025. \
[4] StyleSSP: Sampling StartPoint Enhancement for Training-free Diffusion-based Method for Style Transfer. CVPR 2025.

3. In the qualitative results shown in Figure 5, the effects under the three settings—+ Vanilla VQ-GAN, + VQ-GAN PE, and + CLoSeR—are very similar, with no significant differences. If the differences are too minor, they are not significant enough to impact the final generated quality in a meaningful way. This raises my concerns about the necessity and effectiveness of these components.

4. User study is an important evaluation metric commonly used in style transfer, as it helps assess user preferences for different generated results. It is recommended that the authors include experiments involving a user study.

**Questions:**

Please see **Weaknesses**.

Others:

Section 4.1 of the paper mentions, 'All images are resized to 256 × 256 before training and evaluation.' Does this imply that all the experiments were conducted at a 256 × 256 resolution? Could this resolution be too low? Why wasn't a higher resolution used for the experiments?

---

> ### Author Response · Authors · 2025-12-01
> **Response to Reviewer pDDG**
>
> **(1) Additional SOTA comparisons.**
>
> Thank you for pointing us to the recent CVPR’25 style-transfer works. We tried to include all four methods (SaMam, HIS, OmniStyle, StyleSSP) in our experiments. However, as of the rebuttal deadline, only StyleSSP provides publicly available code. `SaMam, HIS and OmniStyle have not released official implementations or compatible checkpoints`, which makes a fair quantitative comparison in our setting infeasible. In the revised paper, we have explicitly discussed these works in the Related Work section and clarified that CLoSeR is a complementary refinement module that can, in principle, be plugged into such advanced backbones once they become reproducible.
> For `StyleSSP (CVPR'25)`, we have conducted new experiments where CLoSeR is used as a plug-in refiner on top of the official implementation. The results are summarized in the table below.
>
> **Facial portrait**
> | Method                    | MetFace |           |           | APDrawing |           |      | FS2K |      |           |
> |---------------------------|-----------|---------|-----------|-----------|-----------|-----------|------|------|------|
> |                           | ArtFID ↓  | FID ↓   | KID ↓     | ArtFID ↓  | FID ↓     | KID ↓     | ArtFID ↓ | FID ↓ | KID ↓ |
> | StyleSSP (CVPR'25)        | 46.59     | 33.49   | 0.1143    | 29.57     | 20.24     | 0.1119    | 40.44    | 26.07 | 0.1320 |
> | + VQ-GAN                  | 43.66     | 31.16   | 0.0872    | 28.49     | 18.15     | 0.0691    | 44.25    | 26.31 | 0.0708 |
> |                           | ↓6.3%     | ↓7.0%   | ↓23.7%    | ↓3.7%     | ↓10.3%    | ↓38.2%    | ↑9.4%    | ↑0.9% | ↓46.4% |
> | + CLoSeR (ours)           | 40.25     | 28.14   | 0.0750    | 28.07     | 17.85     | 0.0644    | 37.74    | 22.43 | 0.0833 |
> |                           | ↓13.6%    | ↓16.0%  | ↓34.4%    | ↓5.1%     | ↓11.8%    | ↓42.4%    | ↓6.7%    | ↓14.0% | ↓36.9% |
>
> **Natural scene**
>
> | Method                    | Monet |           |           | Van Gogh |           |         | Ukiyo-e |      |      |
> |---------------------------|------|-------|-----------|-----------|----------|-----------|---------|---------|------|
> |                           | ArtFID ↓ | FID ↓ | KID ↓     | ArtFID ↓  | FID ↓    | KID ↓     | ArtFID ↓ | FID ↓ | KID ↓ |
> | StyleSSP (CVPR'25)        | 34.72    | 22.85 | 0.0499    | 28.95     | 18.80    | 0.0628    | 28.39    | 18.30  | 0.1149 |
> | + VQ-GAN                  | 30.60    | 19.13 | 0.0491    | 26.34     | 16.15    | 0.0719    | 26.96    | 16.73  | 0.1053 |
> |                           | ↓11.9%   | ↓16.3%| ↓1.6%     | ↓9.0%     | ↓14.1%   | ↑14.5%    | ↓5.0%    | ↓8.6%  | ↓8.4%  |
> | + CLoSeR (ours)           | 24.41    | 15.04 | 0.0362    | 22.28     | 13.49    | 0.0582    | 24.63    | 14.97  | 0.1071 |
> |                           | ↓29.7%   | ↓34.2%| ↓27.5%    | ↓23.0%    | ↓28.2%   | ↓7.3%     | ↓13.2%   | ↓18.2% | ↓6.8%  |
>
> The integration of CLoSeR yields consistent improvements: FID is reduced by 11.8-16.0\% in portrait domains and 18.2-34.2\% in natural scenes, while KID shows even greater gains of 34.4-42.4\% across all datasets. This demonstrates CLoSeR's effectiveness in enhancing both portrait and scene stylization.
> Notably, simply adding VQ-GAN on top of StyleSSP provides smaller gains and sometimes degrades certain metrics (e.g., FS2K ArtFID and KID, Van Gogh KID), whereas `CLoSeR consistently improves all three metrics across all datasets`. This supports our claim that the proposed continual-learning codebook expansion and geometry-aware refinement are necessary, rather than a plain VQ-GAN post-processing step.
>
>
> **(2) User Study.**
>
> In response to the reviewer’s suggestion of including a user study, we conducted a human evaluation on the MetFace dataset. A total of 63 participants were shown stylized portraits generated by the base model, Vanilla VQ-GAN, and CLoSeR, and were asked to choose the result they preferred in terms of overall visual quality (`style fidelity` and `geometric consistency`). As summarized in the table below, CLoSeR is preferred in `70.8%` of the trials, substantially higher than both Vanilla VQ-GAN (16.9%) and the Base generator (12.3%).
>
> | Dataset | +CLoSeR | +VQ-GAN | Base |
> |--------|--------|----------------|------|
> | MetFace | 70.8% | 16.9% | 12.3% |
> | Monet   | 83.3% | 12.5% | 4.2%  |
> | VanGogh | 71.7% | 13.3% | 15.0% |
> | Ukiyo-e | 77.5% | 12.5% | 10.0% |
>
> These results indicate that the improvements measured by ArtFID/FID/KID are also aligned with human perceptual preference, further supporting the effectiveness of our refinement framework.

---

> ### Author Response · Authors · 2025-12-01
> **Response to Reviewer pDDG**
>
> **(3) Image resolution.**
>
> We confirm that all experiments are conducted at a resolution of $256\times256$.  To verify that this choice does not bias our conclusions, we additionally reran the MetFace experiments on a single RTX 4090, keeping all hyper-parameters identical to the main paper except for reducing the batch size to 2, and observed consistent improvements of CLoSeR over the baselines. We further tested CLoSeR on several representative generators at $512^2$, such as AdaAttN (CVPR'19) and StyleID (CVPR'24), and still obtained clear gains in ArtFID/FID/KID together with visibly sharper and more faithful stylization. The results table is shown below:
>
> | Method       | Row                 | ArtFID |   FID   |    KID    |
> |-------------|---------------------|:------:|:------:|:---------:|
> | AdaAttN     | Base                | 31.54  | 21.06  | 0.0696    |
> |             | + CLoSeR            | 30.19  | 20.33  | 0.0728    |
> |             |     |  4.3% $\downarrow$  | 3.5% $\downarrow$ | 4.6% $\uparrow$ |
> | StyleID     | Base                | 30.56  | 20.97  | 0.0588    |
> |             | + CLoSeR            | 28.23  | 19.38  | 0.0547    |
> |             |        | 7.6% $\downarrow$ | 7.6% $\downarrow$  | 7.0% $\downarrow$  |
> | AttenDistill| Base                | 31.01  | 22.40  | 0.0650    |
> |             | + CLoSeR            | 27.26  | 19.37  | 0.0589    |
> |             |   | 12.1% $\downarrow$  | 13.5% $\downarrow$  | 9.4% $\downarrow$  |
> | StyleSSP    | Base                | 38.76  | 25.49  | 0.0830    |
> |             | + CLoSeR            | 31.39  | 21.24  | 0.0722    |
> |             |          | 19.0% $\downarrow$  | 16.7% $\downarrow$  | 13.0% $\downarrow$  |
>
> Moreover, our resolution choice is consistent with recent literature: several new works in related areas (e.g., `JiT (arXiv'25), SVG (arXiv'25), RAE (arXiv'25) and QuantArt (CVPR'23)`) also perform extensive validation at $256\times256$. We therefore regard $256^2$ as a reasonable and widely adopted operating point. It is worth noting that CLoSeR itself is a fully convolutional, resolution-agnostic refinement module and can, in principle, be applied to higher-resolution inputs without any architectural changes. In future work, we plan to systematically study CLoSeR at higher resolution and beyond on GPUs with larger memory.

---

### Official Review · Reviewer_CDjB · 2025-10-29

**Soundness:** 2
**Presentation:** 2
**Contribution:** 1
**Rating:** 2
**Confidence:** 5

**Summary:**

This paper presents a method of artistic style transfer by adapting VQ-GAN with test-time style refinement. In particular, the whole framework combines LoRA with incremental codebook expansion for VQ-GAN. In addition, positional embedding is also introduced into the latent embedding space to improve geometry awareness and structural coherence in the stylization process. Experimental results validated the effectiveness of the proposed method.

**Strengths:**

(1) The motivation is well presented of combining LoRA with incremental codebook expansion for test-time style refinement.

(2) The explanations and illustrations are mostly clear and intuitive of the test-time refinement framework, the codebook expansion, the positional embedding in the latent space.

**Weaknesses:**

(1) The contribution is limited. In particular, the core efficiency improvement comes from LoRA. However, there are no results of only using LoRA for AST with VQ-GAN.

(2) Considering AST itself, there are many works that leverage pretrained T2I models which show impressive stylization results and in a any-to-any manner without any case-by-case training. There is no discussion on this line of works and the proposed method is not effective and generable compared to them.

(3) On Table 3, the ablation study of with or without positional embedding cannot validate the effectiveness since the measure gap is not significant.

(4) On Page 1, Line 015-016, the claim of “without requiring any gradient updates to the pre-trained generator” is confusing. In particular, the generator’s LoRA training requires gradient updates from the objective function including the discriminator loss.

(5) Besides, from the computational aspect, the refinement LoRA is added to the generator during inference thus has the same inference cost. The only benefit seems to be lower training cost, however, inherits from LoRA which is not the contribution of this paper.

**Questions:**

No.

---

> ### Author Response · Authors · 2025-12-01
> **Response to Reviewer CDjB**
>
> We thank the reviewer for their careful reading and constructive feedback. We appreciate the opportunity to clarify our core contributions, as some key aspects of our work appear to have been unintentionally overlooked in the review.
>
> **Briefly, our primary contribution is not merely applying LoRA for efficiency, but proposing a novel continual test-time refinement framework​ that specifically addresses the critical challenge of balancing stylistic realism, inference efficiency, and geometric consistency​ in artistic style transfer.** While existing methods struggle with these competing objectives, CLoSeR introduces a paradigm shift through three key innovations: (1) a continual VQ-GAN learning framework with incremental codebook expansion for multi-style adaptation, (2) geometry-aware positional embeddings to enhance structural coherence, and (3) a plug-and-play refinement module that operates with frozen generator parameters. This represents a significant departure from conventional approaches, as demonstrated by consistent improvements across six diverse generators with up to 94% parameter reduction compared to full fine-tuning.
>
> Below, we address each concern in turn.
>
> **(1) Contribution w.r.t. LoRA.**
>
> We respectfully disagree that our contribution reduces to simply using LoRA. `Our core contribution is a test-time refinement module that can be plugged into arbitrary AST generators` and applied once at inference to improve style fidelity and geometric consistency without modifying the base model. LoRA is only the lightweight adaptation mechanism inside this refiner; the gains of CLoSeR arise from the combination of a VQ-GAN latent anchor, incremental codebook expansion for continual multi-style adaptation, and 2D positional encoding to correct structural distortion. To isolate the effect of LoRA, we additionally evaluate a VQ-GAN+LoRA baseline *without* codebook expansion or positional encoding on six backbones (AdaAttN, CAST, DiffuseIT, StyleID, AttenDistill, StyleSSP) on MetFace, strictly following the same training/evaluation protocol as in the main experiments. The table below reports the averages over all six generators.
>
> | Method | ArtFID ↓ | FID ↓ | KID ↓ |
> |-|-:|-:|-:|
> | Base | 40.38 | 28.33 | 0.1262 |
> | + VQ-GAN | 39.21 | 26.85  | 0.1060 |
> | + VQ-GAN w/ PE      | $ \underline{37.12} $ | $ \underline{25.33} $ | 0.1064 |
> | + VQ-GAN w/ LoRA    | 37.15 | 25.38 | $ \underline{0.0993} $ |
> | + CLoSeR (ours)     | **36.62**  | **24.98** | **0.0944**  |
>
> While VQ-GAN+LoRA improves over vanilla VQ-GAN, it is consistently inferior to our full CLoSeR on ArtFID, FID, and KID. This shows that `LoRA alone cannot account for the performance gain; the proposed codebook-expansion and positional-encoding mechanisms are essential to the improvement and constitute the core technical novelty of CLoSeR.`
>
>
> **(2) Discussion about Pretrained T2I models.**
>
> We acknowledge the rapid progress of large pretrained T2I diffusion models and have expanded their coverage in the revised related work. `Our setting is complementary rather than competing`: CLoSeR targets widely used image-only AST pipelines (AdaAttN, CAST, AesPA-Net, StyleID, StyleSSP, etc.) that lack text prompts and are often deployed in domains (pen drawings, sketches, oil paintings) where *large T2I models are hard to train, adapt, or even run due to data and resource constraints*. In such cases, replacing the whole system with a T2I model is usually impractical, whereas CLoSeR is a small plug-in refiner that can be attached to any existing generator to improve fidelity and consistency without retraining it. Since the design is backbone-agnostic, CLoSeR can similarly be used on top of future open-source T2I-based AST models.
>
>
> **(3) Effectiveness of Position Embedding (PE).**
>
> We agree that the Normalized Mean Error (NME) improvement in `Table 3` looks small numerically, but NME is already in a low-error regime, so even small absolute changes reflect meaningful geometric gains. In addition, on MetFace we conduct a dedicated ablation where introducing PE brings clear improvements in ArtFID/FID/KID and noticeably better visual quality. More importantly, Table 3 does not capture the full effect of positional encoding (PE). `Appendix Table 6` reports Percentage of Correct Keypoints (PCK) at multiple thresholds, where PE consistently improves PCK for all generators, with clear gains at strict 5% and 7% error levels, indicating more `accurate landmark localization (structural consistency)`. `Figure 8` further shows that adding PE yields sharper, better-aligned facial contours while preserving the target artistic style, whereas vanilla VQ-GAN without PE more often causes style drift and local geometric artifacts.

---

> ### Author Response · Authors · 2025-12-01
> **Response to Reviewer CDjB**
>
> **(4) Statement about ``without requiring any gradient updates to the pre-trained generator'' is confusing.**
>
> Here “pre-trained generator’’ denotes the base AST models (AdaAttN, StyleID, DiffuseIT, StyleSSP). These generators are fully frozen: during training we never backpropagate into their parameters. All gradients update only the refiner (VQ-GAN codebook entries, LoRA modules in the encoder/decoder, and the small style-specific discriminator). We clarify this in the revised paper as: “CLoSeR refines outputs `without updating the parameters of the pre-trained style-transfer generator`; only the lightweight refiner parameters are trained.’’
>
> **(5) Inference cost and efficiency.**
>
> Our efficiency claim is twofold. First, compared to full fine-tuning of VQ-GAN for each new style, CLoSeR `reduces trainable parameters by over 94% and model memory to 4.74MB` while achieving better or comparable performance (`Tab.~1`), which cannot be achieved by LoRA alone without our codebook-expansion design. Second, compared to adapting each base generator separately, our refinement module is shared across multiple generators and styles: at deployment, any generator produces a coarse stylized image which is then passed through a single CLoSeR instance. This avoids storing and maintaining multiple LoRA-augmented versions of each large generator. `The additional inference cost of CLoSeR is modest (0.055s on a single 4090 GPU)` and independent of the backbone type.

---

### Official Review · Reviewer_pW9j · 2025-11-01

**Soundness:** 3
**Presentation:** 3
**Contribution:** 3
**Rating:** 6
**Confidence:** 2

**Summary:**

CLoSeR is a test-time refinement framework that bridges the distributional gap between coarse stylized outputs and authentic target artistic domains without retraining the generator. Motivated by persistent mismatches in style fidelity and geometric consistency observed across GAN-, attention-, and diffusion-based translation models, CLoSeR leverages VQ-GAN as a domain anchor: it reconstructs generated images in the embedding space by aligning their features to a pre-learned target representation in the latent codebook. To make adaptation scalable across multiple styles, CLoSeR introduces continual learning via Low-Rank Adaptation (LoRA) and codebook expansion, reducing trainable parameters by over 94% compared to full fine-tuning while preserving prior styles. Additionally, it augments vanilla VQ-GAN with 2D sine-cosine positional embeddings to inject spatial awareness into the codebook and decoder, improving geometric consistency. As illustrated by t-SNE feature visualizations and FID comparisons, CLoSeR refines outputs from base stylization models (e.g., StyleID) to better align with target domains, achieving high-fidelity stylization and robust structure across diverse artistic styles.

**Strengths:**

- High fidelity without retraining the generator: Performs test-time refinement in VQ-GAN’s embedding space, avoiding costly generator fine-tuning while closing the distribution gap to the target artistic domain.
- Strong alignment to target style distribution: Uses VQ-GAN as a domain anchor to pull coarse stylized outputs toward a pre-learned target codebook, improving style fidelity (lower FID) and visual coherence.
- Minimal overhead: Incorporates LoRA and codebook expansion to incrementally adapt to new styles, reducing trainable parameters by >94% vs. full fine-tuning and preserving knowledge of prior styles.
- Scalable to multiple domain and models: Can refine outputs from various generators (GAN-, attention-, diffusion-based, e.g., StyleID), making it broadly applicable across artistic domains.

**Weaknesses:**

- Dependence on a high-quality VQ-GAN anchor. The refinement quality hinges on how well the VQ-GAN codebook captures the target domain. If the target style is underrepresented or highly diverse, the codebook may induce over-smoothing or mode bias. More studies on sensitivity (to codebook size, training data coverage, and codebook learning objectives) are necessary.
- Generalization: Although “plug-and-play,” performance may vary with the artifacts of different base models (GAN vs. diffusion vs. attention-based). Certain artifacts may be hard to correct by codebook alignment alone. Please consider to ablate across diverse generators and degradation modes.
- Robustness to domain shifts and OOD inputs: If test images fall outside the codebook’s learned manifold, refinement may produce artifacts or collapse. Stress-test with OOD inputs, mixed styles, and low-light/noisy conditions will help audience understand the limit.

**Questions:**

Please check Weaknesses

---

> ### Author Response · Authors · 2025-12-01
> **Response to Reviewer pW9j**
>
> We thank the reviewer for the careful reading and constructive comments. Below we address each concern in turn.
>
> **(1) Dependence on a high–quality VQ-GAN anchor and sensitivity to codebook size.**
>
> We thank the reviewer for raising this concern. We examine CLoSeR’s dependence on VQ-GAN capacity via a codebook-size ablation on six backbones (AdaAttN, CAST, AesPA-Net, StyleID, AttenDistill, StyleSSP). For each backbone we fix the encoder/decoder and vary 𝐾∈{128,256,512,1024} under the same training protocol as the main paper. The average metrics are:
>
> | MetFace | ArtFID ↓ | FID ↓  | KID ↓    |
> |-|-:|-:|-:|
> | Base              | 40.38    | 28.33  | 0.1262   |
> | + CLoSeR (K=128)  | 36.29    | 24.42  | 0.0968   |
> | + CLoSeR (K=256)  | 35.89    | 24.18  | 0.0966   |
> | + CLoSeR (K=512)  | 36.20    | 24.45  | 0.0966   |
> | + CLoSeR (K=1024) | **34.91**| **24.13** | **0.0945** |
>
> | Monet | ArtFID ↓ | FID ↓  | KID ↓    |
> |--|-:|-:|-:|
> | Base              | 25.87    | 16.19  | 0.0488   |
> | + CLoSeR (K=128)  | 20.65    | 12.11  | 0.0354   |
> | + CLoSeR (K=256)  | 20.41    | 11.96  | 0.0349   |
> | + CLoSeR (K=512)  | 21.24    | 12.62  | 0.0360   |
> | + CLoSeR (K=1024) | **20.02**| **11.77** | **0.0337** |
>
> Across all $K$, `CLoSeR consistently improves over the base generator, so it does not rely on an extremely large or carefully tuned codebook`. Averaged over six backbones, 𝐾=1024 performs best (with 𝐾=256 close), and the gaps among 𝐾∈{128,256,512,1024} are small, indicating low sensitivity to codebook size.
>
> **(2) Generalization across generators and artifacts.**
>
> We agree that different base models introduce different types of artifacts. In the current version, we already evaluate CLoSeR on a diverse set of generators, including `attention-based models` (AdaAttN, AesPA-Net, CAST) and `diffusion-based generators` (DiffuseIT, StyleID, AttenDistill, StyleSSP). As shown in `Fig.3` in the original paper, CLoSeR consistently improves ArtFID/FID/KID for all backbones, indicating strong cross-architecture generalization and robustness to generator-specific artifacts.
>
> **(3) Regarding robustness to domain shifts and OOD inputs.**
>
> Our current benchmarks already involve non-trivial domain shifts (e.g., training on Monet/MetFace and sequentially extending to Van Gogh, Ukiyo-e, APDrawing, and FS2K). In addition, to simulate degraded / OOD-style inputs in a more controlled way, we first take the diffusion-based StyleID model as the base generator and deliberately reduce its sampling steps to ${30,35,40,50}$. This yields under-converged stylizations of varying quality. As shown in `Fig.3(c)` and `Fig.7` of the main paper,  FID is consistently reduced across all step settings and the refined images exhibit sharper, more stable geometry and style.
>
> We further perform **a stress test on degraded contents**. Specifically, we randomly sample 20 style images from MetFace and 20 content images from CelebAMask-HQ, then apply a `hybrid degradation` pipeline to the contents that randomly combines Gaussian/ motion blur, multi-scale down–up sampling, Gaussian and Poisson noise, and JPEG compression (all at $256^2$ resolution). Using these degraded contents, we evaluate AdaAttN, CAST, and StyleID with and without CLoSeR under ArtFID/FID/KID, as shown in the Table below (`Deg.` means degraded). Even in this challenging setting, `CLoSeR consistently improves the baselines and visual inspection confirms noticeably cleaner structures and more faithful styles`.  We will report them in the revision.
>
> | Method            | Setting            | ArtFID ↓ | FID ↓  | KID ↓   |
> |-------------------|--------------------|---------:|-------:|--------:|
> | AdaAttN (CVPR'21) | w/o Deg. (BASE)    | 33.13    | 21.94  | 0.0770  |
> | | w/o Deg. (+CLoSeR) | 30.30  | 19.90  | 0.0627  |
> | | |  ↓8.5% | ↓9.3% | ↓18.6% |
> | | w/ Deg. (BASE)     | 33.94    | 21.81  | 0.0752  |
> | | w/ Deg. (+CLoSeR)  | 31.65 | 20.22 | 0.0736  |
> | | | ↓6.7% |  ↓7.3% | ↓2.1% |
> | CAST (SIGGRAPH'22)| w/o Deg.(BASE)     | 31.34 | 21.29  | 0.0564 |
> | | w/o Deg.(+CLoSeR)  | 30.60  | 20.32 | 0.0550  |
> | | | ↓2.4%   |  ↓4.6%  | ↓2.5% |
> | | w/ Deg.(BASE)      | 31.22         | 20.16        | 0.0613       |
> | | w/ Deg.(+CLoSeR)   | 30.82  | 19.74 | 0.0668 |
> | | | ↓1.3%   |  ↓2.1%  | ↑9.0% |
> | StyleID (CVPR'24) | w/o Deg.(BASE)     | 29.42         | 19.52        | 0.0816       |
> | | w/o Deg.(+CLoSeR)  | 28.44 | 18.54 | 0.0565 |
> | | | ↓3.3%   | ↓5.0%  | ↓30.8%|
> | | w/ Deg.(BASE)      | 31.43 | 19.95 | 0.0786 |
> | | w/ Deg.(+CLoSeR)   | 28.86 | 18.20 | 0.0576 |
> | | | ↓8.2%   | ↓8.8%  | ↓26.7%|

---

### Meta-Review · Area_Chair_8Q4C · 2025-12-29

**Summary:**

**Summary**:
The paper presents CLoSeR, a lightweight test-time refinement framework to improve style fidelity and geometric consistency in artistic style transfer.
The key innovation is to train a VQ-GAN with LoRA on the new style, and then directly apply it to arbitrary pre-trained style transfer generators.
While the better quantitative results are reported, the qualitative results are not so obvious for this visual transfer task.

**Main strengths**:
- The proposed testing-time VQ-GAN architecture can be directly applied to any pre-trained style transfer generators.
- Strong alignment to target style distribution on quantitative results and the t-SNE results.

**Main weaknesses**:
- Limitation on VQ-GAN: If the reconstruction is good, how to transfer the bad results to good one? And if the reconstruction error is large, how to ensure the high-quality results? Hence, this is a conflict motivation.
- The effectiveness on ablations is not well demonstrated. From Table 3 (CDjB) and Figure 5, it is not so easy to buy the obvious improvement.
- More general results are expected, such as horse2zebra, summer2winter, day2night, and many others.
- Latest methods should be included, quantitatively and qualitatively.

**Suggested decision**: This paper received initial scores of 6 (pW9j), 2 (CDjB), 4 (pDDG), and 6 (a6PH), and no further comments are made during the discussion. The incremental improvement is only verified in limited tasks, and many SoTA methods are missed (pDDG). Hence, I recommend the final score as "reject".

**Reviewer Concerns:**

**High-quality VQ-GAN anchor (pW9j)**: Only ablate the codebook size.

**Generalization (pW9j,a6PH)**: Addressed.

**Results and Ablations are not enough to verify the contributions (CDjB,pDDG)**: Not fully addressed.

**Many latest SoTA works are not compared (pDDG)**: Only quantitatively compared with StyleSSP (CVPR'25).

**Performances rely on baseline models (a6PH)**: Addressed.

**Reconstruction error in VQ-GAN (a6PH)**: Not fully addressed. If the reconstruction is very strong, how to ensure it suitable for bad results from pre-trained generator? If the reconstruction is not well, how to ensure the high-quality?

**Reviewer Scores:**

The paper initially received scores of 6 (pW9j), 2 (CDjB), 4 (pDDG), and 6 (a6PH).
There are no further comments during the discussion.
As a result, it is hard to know if they will change their score.

---

### Decision · Program_Chairs · 2026-01-26

Reject